# COINPRESS: Practical Private Mean and Covariance Estimation*

**Sourav Biswas**
Cheriton School of Computer Science
University of Waterloo
s23biswa@uwaterloo.ca

**Yihe Dong**
Microsoft
yihdong@microsoft.com

**Gautam Kamath**
Cheriton School of Computer Science
University of Waterloo
g@csail.mit.edu

**Jonathan Ullman**
Khoury College of Computer Sciences
Northeastern University
jullman@ccs.neu.edu

## Abstract

We present simple differentially private estimators for the mean and covariance of multivariate sub-Gaussian data that are accurate at small sample sizes. We demonstrate the effectiveness of our algorithms both theoretically and empirically using synthetic and real-world datasets—showing that their asymptotic error rates match the state-of-the-art theoretical bounds, and that they concretely outperform all previous methods. Specifically, previous estimators either have weak empirical accuracy at small sample sizes, perform poorly for multivariate data, or require the user to provide strong a priori estimates for the parameters.

## 1   Introduction

One of the most basic problems in statistics and machine learning is to estimate the mean and covariance of a distribution based on i.i.d. samples. The optimal solutions to these problems are folklore—simply output the empirical mean and covariance of the samples. However, this solution is not suitable when the samples consist of sensitive, private information belonging to individuals, as it has been shown repeatedly that even releasing just the empirical mean can reveal this sensitive information [13, 23, 8, 18, 17]. Thus, we need estimators that are not only accurate with respect to the underlying distribution, but also protect the *privacy* of the individuals represented in the sample.

The most widely accepted solution to individual privacy in statistics and machine learning is *differential privacy* (DP) [15], which provides a strong guarantee of individual privacy by ensuring that no individual has a significant influence on the learned parameters. A large body of work now shows that, in principle, nearly every statistical task can be solved privately, and differential privacy is now being deployed by Apple [11], Google [20, 3], Microsoft [12], and the US Census Bureau [10].

Differential privacy requires adding random noise to some stage of the estimator, and this noise might increase the error of the final estimate. Typically, the amount of noise vanishes as the sample size $n$ grows, and one can often show that as $n \to \infty$, the additional error due to privacy vanishes faster than the sampling error of the estimator, making differential privacy highly practical for large samples.

However, differential privacy is often difficult to achieve for small datasets, or when we want to look at some small subpopulation within a large dataset. Thus, a recent trend has been to focus on simple,

widely used estimation tasks, and design estimators with good concrete performance at small sample sizes. Most relevant to our work, Karwa and Vadhan [27] and Du, Foot, Moniot, Bray, and Groce [14] give practical mean and variance estimators for univariate Gaussian data. However, as we show, these methods do not scale well to the more challenging multivariate setting.

## 1.1 Contributions

In this work we give simple, practical estimators for the mean and covariance of *multivariate* sub-Gaussian data.[2] We validate our estimators theoretically and empirically. On the theoretical side, we show that our estimators match the state-of-the-art asymptotic bounds for private sub-Gaussian mean and covariance estimation [24]. On the empirical side, we give an extensive evaluation with synthetic data, as well as a demonstration on a real-world dataset. We show that our estimators have error comparable to that of the non-private empirical mean and covariance at samples sizes on the order of 1,000. Our mean estimator also improves over the state-of-the-art method [14], which was developed for univariate data but can be applied coordinate-wise to estimate multivariate data.

We highlight that, like most private estimators, we require the user to input some a priori knowledge of the mean and covariance. For mean estimation, we require the mean lives in a specified ball of radius $R$, and for covariance estimation we require that the covariance matrix can be sandwiched spectrally between $A$ and $KA$ for some matrix $A$. Some a priori boundedness is *necessary* for algorithms like ours that satisfy concentrated DP [16, 6, 29, 4], or satisfy pure DP.[3] We show that our estimator is practical when these parameters are taken to be extremely large, meaning the user only needs a very weak prior. Also, for simplicity, we focus on Gaussian data, but using both heavy-tailed synthetic data and real-world data, we show that our method remains useful beyond Gaussian data. Note that some restriction on the decay of the tails is necessary in the worst case [26].

**Approach.** At a high-level, our estimators iteratively refine an estimate of the parameters. For mean estimation, we start with some (potentially large) ball $B_1$ of radius $R_1$ that contains most of the mass of the probability distribution. We use this ball to get a naïve estimate: clip the data to the ball, then obtain an estimate of the mean by adding noise to the empirical mean of the clipped data, with magnitude proportional to $R_1/n$. Using this estimate, and knowledge of how we obtained it, we can draw a (hopefully smaller) ball $B_2$ of radius $R_2$ that contains most of the mass and then repeat. After a few iterations, we will have some ball $B_t$ of radius $R_t$ that tightly contains most of the datapoints, and use this to make an accurate final private estimate of the mean with noise proportional to $R_t/n$. Our covariance estimation uses the same iterative approach, although the geometry is significantly more subtle.

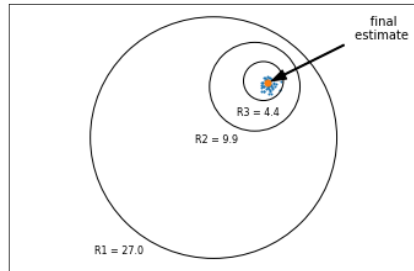

Figure 1: Visualizing a run of the mean estimator with $n = 160, \rho = 0.1, t = 3$

## 1.2 Problem Formulation

We now detail the problem we consider in this work. We are given samples $X = (X_1, \ldots, X_n) \subseteq \mathbb{R}^d$ where each $X_i$ represents some individual's sensitive data. We would like an estimator $M(X)$ that is *private* for the individuals in the sample, and also *accurate* in that when $X$ consists of i.i.d. samples from some distribution $P$, then $M(X)$ estimates the mean and covariance of $P$ with small error. Note that accuracy relies on distributional assumptions, but the privacy guarantee is worst-case. For privacy, we require that $M$ is insensitive to any one datapoint in the following sense: We say that two samples $X, X' \subseteq \mathbb{R}^d$ of size $n$ are *neighboring* if they differ on at most one datapoint.[4] Informally, $M$ is *differentially private* [15] if the distributions $M(X)$ and $M(X')$ are similar for every pair of

neighboring datasets $X, X'$.[5] In this work, we adopt *concentrated differential privacy (zCDP)* [16, 6].

**Definition 1.1.** $M(X)$ *satisfies $\rho$-zCDP if for every pair of neighboring samples $X, X'$ of size $n$, and every $\alpha \in (1, \infty)$, $D_\alpha(M(X)\|M(X')) \le \rho\alpha$, where $D_\alpha$ is the Rényi divergence of order $\alpha$.*

This formulation sits in between general $(\varepsilon, \delta)$-differential privacy and the special case of $(\varepsilon, 0)$-differential privacy,[6] and better captures the privacy cost of private algorithms in high dimension [17].

To formulate the accuracy of our mechanism, we posit that $X$ is sampled i.i.d. from some distribution $P$, and our goal is to estimate the mean $\mu \in \mathbb{R}^d$ and covariance $\Sigma \in \mathbb{R}^{d \times d}$ with $\mu = \mathbb{E}_{x \sim P}[x]$ and $\Sigma = \mathbb{E}_{x \sim P}[(x - \mu)^T(x - \mu)]$. We assume that our algorithms are given some a priori estimate of the mean in the form of a radius $R$ such that $\|\mu\|_2 \le R$ and some a priori estimate of the covariance in the form of $K$ such that $I \preceq \Sigma \preceq KI$ (equivalently all singular values of $\Sigma$ lie in $[1, K]$).

We measure the error in *Mahalanobis* distance $\|\cdot\|_\Sigma$, which compares the error to the covariance of the distribution, and has the benefit of being invariant under affine transformations. Specifically,

$$\|\hat{\mu} - \mu\|_\Sigma = \|\Sigma^{-1/2}(\hat{\mu} - \mu)\|_2 \quad \text{and} \quad \|\hat{\Sigma} - \Sigma\|_\Sigma = \|\Sigma^{-1/2}\hat{\Sigma}\Sigma^{-1/2} - I\|_F.$$

For any distribution, the *empirical mean* $\hat{\mu}$ and *empirical covariance* $\hat{\Sigma}$ satisfy $\mathbb{E}[\|\hat{\mu} - \mu\|_\Sigma] \le \sqrt{d/n}$ and $\mathbb{E}[\|\hat{\Sigma} - \Sigma\|_\Sigma] \le \sqrt{d^2/n}$, and these estimators are minimax optimal. Our goal is to obtain estimators that have similar accuracy to the empirical mean and covariance. The folklore naïve estimators (see e.g. [27, 24]) for mean and covariance based on clipping the data to an appropriate ball and adding carefully calibrated noise to the empirical mean and covariance would guarantee

$$\mathbb{E}[\|\hat{\mu} - \mu\|_\Sigma] \le \sqrt{\frac{d}{n}} + \frac{Rd}{n\sqrt{\rho}} \quad \text{and} \quad \mathbb{E}\Big[\|\hat{\Sigma} - \Sigma\|_\Sigma\Big] \le \sqrt{\frac{d^2}{n}} + \frac{Kd^2}{n\sqrt{\rho}}.$$

The downside of the naïve estimators is that they depend linearly on $R$ and $K$, and thus have large error unless the user has strong a priori knowledge of the mean and covariance. Requiring users to provide such a priori bounds is a major challenge in systems for differentially private analysis [21]. Our estimators have much better concrete and asymptotic dependence on these parameters.

**Related Work.** The most relevant line of work is that initiated by Karwa and Vadhan [27], studying private parameter estimation for Gaussian data focused on important issues for practice like dealing with weak a priori bounds on the parameters. Later works studied the multivariate setting [24, 9, 25] and estimation under weaker moment assumptions [7, 26], though these investigations are primarily theoretical. Our algorithm for covariance estimation can be seen as a simpler and more practical variant of [24]: they iteratively threshold eigenvalues to find directions of high and low variance, whereas we employ a softer method to avoid wasting information. To be explicit: the results of [24] focused on getting algorithms which were theoretically sample near-optimal. However, this is far from designing practical or realizable methods – our work can be seen as a refinement of their results to provide algorithms which achieve reasonable error with minimal hyperparameter tuning. A notable work is [14], providing practical private confidence intervals in the univariate setting. Instead, our investigation is focused on realizable algorithms for the multivariate setting. Several works consider private PCA or covariance estimation [19, 22, 2], though, unlike our work, these methods assume strong a priori bounds on the covariance. Additional related work appears in the supplement.

## 2 New Algorithms for Multivariate Gaussian Estimation

We present new algorithms for Gaussian parameter estimation (Figure 2). While these do not result in improved asymptotic sample complexity bounds, they are much more practical in the multivariate setting. In particular, they will avoid the curse of dimensionality incurred by multivariate histograms, but also eschew many of the hyperparameters that arise in previous methods [24]. Note that our algorithms with $t = 1$ precisely describe the naïve method that was informally outlined in Section 1.2. We describe the algorithms and sketch ideas behind the proofs, which appear in the supplement (along with simpler univariate algorithms in the same style). Our core primitive is the Gaussian mechanism.

<div style="border:1px solid">

**Input:** $X_{1\ldots n} \sim N(\mu, I_{d\times d})$, $B_2(c,r)$ containing $\mu$, $t \in \mathbb{N}^+$, $\rho_{1\ldots t}, \beta > 0$
**Output:** A $(\sum_t \rho_t)$-zCDP estimate of $\mu$
MVMREC$(X_{1\ldots n}, c, r, t, \rho_{1\ldots t}, \beta)$:

1. Let $(c_0, r_0) = (c, r)$

2. For $i = 1, \ldots, t-1$:
$(c_i, r_i) = \text{MVM}(X_{1\ldots n}, c_{i-1}, r_{i-1}, \rho_t, \frac{\beta}{4(t-1)})$

3. $(c_t, r_t) = \text{MVM}(X_{1\ldots n}, c_{t-1}, r_{t-1}, \rho_t, \frac{\beta}{4})$

4. Return $c_t$

</div>

<div style="border:1px solid">

**Input:** $X_{1\ldots n} \sim N(\mu, I_{d\times d})$, $B_2(c,r)$ containing $\mu$, $\rho_s, \beta_s > 0$
**Output:** A $\rho_2$-zCDP ball $B_2(c', r')$
MVM$(X_{1\ldots n}, c, r, \rho_s, \beta_s)$:

1. Let $\gamma = \sqrt{d + 2\sqrt{d\log(\frac{n}{\beta_s})} + 2\log(\frac{n}{\beta_s})}$

2. Project each $X_i$ into $B_2(c, r+\gamma)$

3. Let $Z = \frac{1}{n}\sum_i X_i + Y$, $Y \sim N(0, \frac{2(r+\gamma)^2}{n^2 \rho_s}I)$

4. Let $c' = Z$, $r' = \gamma\sqrt{\frac{1}{n} + \frac{2(r+\gamma)^2}{n^2 \rho_s}}$

5. Return $(c', r')$

</div>

Figure 2: Mean Estimation Algorithms

**Lemma 2.1.** *Let $f$ be an $\mathbb{R}^d$-valued function with $\ell_2$-sensitivity $\Delta_{f,2} = \max_{X \sim X' \in \mathcal{X}^n} \|f(X) - f(X')\|_2$, where $X \sim X'$ denotes that $X$ and $X'$ are neighboring databases, differing in at most one entry. Then the Gaussian mechanism $M_f(X) = f(X) + N(0, (\frac{\Delta_{f,2}}{\sqrt{2\rho}})^2 \cdot I)$ satisfies $\rho$-zCDP.*

## 2.1 Multivariate Private Mean Estimation

We first present our multivariate private mean estimation algorithm MVMREC. This is an iterative algorithm, which maintains a confidence ball that contains the true mean with high probability. For ease of presentation, we state the algorithm for a Gaussian with identity covariance, though a rescaling argument allows it to work for an arbitrary known covariance $\Sigma$.

MVMREC calls MVM $t-1$ times, each time with a new $\ell_2$-ball $B_2(c_i, r_i)$ centered at $c_i$ with radius $r_i$. We desire that each invocation is such that $\mu \in B_2(c_i, r_i)$, and the $r_i$'s should decrease rapidly, so that we quickly converge to a fairly small ball which contains the mean. Our goal will be to acquire a small enough radius. With this in hand, we can run the naïve algorithm which clips the data and applies the Gaussian mechanism. With large enough $n$, this will have the desired accuracy.

It remains to reason about MVM. We need to argue: a) privacy; b) accuracy: given $B_2(c,r) \ni \mu$, it is likely to output $B_2(c', r') \ni \mu$; and c) progress: the radius $r'$ output is much smaller than the $r$ input. The algorithm first chooses some $\gamma$ and clips the data to $B_2(c, r+\gamma)$. $\gamma$ is chosen based on Gaussian tail bounds such that if $\mu \in B_2(c,r)$, then none of the points will be affected by this operation. This bounds the sensitivity of the empirical mean, and applying the Gaussian mechanism guarantees privacy. While the noised mean will serve as a point estimate $c'$, we actually have more: again using Gaussian tail bounds on the data combined with the added noise, we can define a radius $r'$ such that $\mu \in B_2(c', r')$, establishing accuracy. Finally, a large enough $n$ will ensure $r' < r/2$, establishing progress. Since each step reduces our radius $r_i$ by a constant factor, setting $t = O(\log R)$ will reduce the initial radius from $R$ to $O(1)$ as desired. Formalizing this gives the following theorem.

**Theorem 2.2.** *MVMREC is $(\sum_{i=1}^t \rho_i)$-zCDP. Furthermore, suppose $X_1, \ldots, X_n$ are samples from $N(\mu, I)$ with $\mu$ contained in the ball $B_2(C, R)$, and $n = \tilde{\Omega}((\frac{d}{\alpha^2} + \frac{d}{\alpha\sqrt{\rho}} + \frac{\sqrt{d\log R}}{\sqrt{\rho}}) \cdot \log\frac{1}{\beta})$. Then MVMREC$(X_1, \ldots, X_n, C, R, t = O(\log R), \frac{\rho}{2(t-1)}, \ldots, \frac{\rho}{2(t-1)}, \frac{\rho}{2}, \beta)$ will return $\hat{\mu}$ such that $\|\mu - \hat{\mu}\|_\Sigma = \|\mu - \hat{\mu}\|_2 \leq \alpha$ with probability at least $1 - \beta$.*

## 2.2 Multivariate Private Covariance Estimation

We describe our multivariate private covariance estimation algorithm, which due to space restrictions, appears in the supplement. The ideas conceptually similar to mean estimation, but subtler due to the more complex geometry. We assume data is drawn from $N(0, \Sigma)$ where $I \preceq \Sigma \preceq KI$ for some known $K$. One can reduce to the zero-mean case by differencing pairs of samples. Further, if we know some PSD matrix $A$ such that $A \preceq \Sigma \preceq KA$ then we can rescale the data by $A^{-1/2}$.

Similar to MVMREC, we repeatedly call a private algorithm that makes a constant-factor progress, and then run the naïve algorithm. Rather than maintaining a ball containing the true mean, we

maintain a pair of ellipsoids which sandwich the true covariance between them (in fact, we rescale at each step so that the outer covariance is always the identity). Once the axes of these ellipsoids are within a constant factor we can apply the naïve algorithm, which is accurate given enough samples.

The step algorithm is similar to before. We first clip the points at a distance based on Gaussian tail bounds with respect to the outer ellipsoid, which is unlikely to affect the dataset when it dominates the true covariance. After this operation, we can show that the sensitivity of an empirical covariance statistic is bounded using the following lemma. [24] proved a similar statement without an explicit constant, but the optimal constant is important in practice.

**Lemma 2.3.** *Consider the function $f(D) = \frac{1}{n} \sum_i D_i D_i^T$, where $\|D_i\|_2^2 \leq \tau$. Then the $\ell_2$-sensitivity of $f$ (i.e., $\max_{D \sim D'} \|f(D) - f(D')\|_F$ where $D \sim D'$ are neighbors) is at most $\frac{\tau \sqrt{2}}{n}$.*

Applying the Gaussian mechanism (à la [19]) in combination with this sensitivity bound, we again get a private point estimate $Z$ for the covariance, and can also derive inner and outer confidence ellipsoids. This time we require more sophisticated tools, including confidence intervals for the spectral norm of both a symmetric Gaussian matrix and the empirical covariance matrix of Gaussian data. Using valid confidence intervals ensures accuracy, and a sufficiently large $n$ again results in a constant factor squeezing of the ellipsoids, guaranteeing progress. This leads to the following theorem.

**Theorem 2.4.** *There exists an algorithm MVCREC which is $(\sum_{i=1}^t \rho_i)$-zCDP. Furthermore, suppose $X_1, \ldots, X_n \sim N(0, \Sigma)$, where $I \preceq \Sigma \preceq KI$, and $n = \tilde{\Omega}((\frac{d^2}{\alpha^2} + \frac{d^2}{\alpha\sqrt{\rho}} + \frac{\sqrt{d^3 \log K}}{\sqrt{\rho}}) \cdot \log \frac{1}{\beta})$. Then running the algorithm MVCREC($X_1, \ldots, X_n, I, K, t = O(\log K), \frac{\rho}{2(t-1)}, \ldots, \frac{\rho}{2(t-1)}, \frac{\rho}{2}, \beta$) will return $\hat{\Sigma}$ such that $\|\hat{\Sigma}^{-1/2} \Sigma \hat{\Sigma}^{-1/2} - I\|_F \leq \alpha$ with probability at least $1 - \beta$.*

## 3  Mean Estimation Experiments

We now present our experimental results on multivariate mean estimation. Additional experiments, and code to reproduce these experiments, appear in the supplement. At a high level, there are two general approaches for the multivariate problem. The first is to solve the univariate problem in each dimension, and combine these results in the natural way. As shown in [24], with an appropriate setting of parameters, an asymptotically optimal algorithm for the univariate problem leads to an asymptotically near-optimal algorithm for the multivariate problem. The other class of approaches is to work directly in the multivariate space, as done in our novel method MVMREC (Figure 2).

We compare the following approaches, labeled with their names as in the legend of our plots: (1) The non-private empirical mean (`Non-private`); (2) Univariate naïve method applied coordinatewise (`Naive coordinatewise`); (3) Karwa-Vadhan [27] applied coordinatewise (KV); (4) SYMQ of Du et al. [14] applied coordinatewise (SYMQ); (5) Multivariate naïve method (MVMREC with $t = 1$) (`t = 1`); (6) MVMREC for various $t > 1$ (`t = ♠`, for integer ♠ > 1).

**Implementation Details.** We use our own implementation of these algorithms except for SYMQ, for which we use the code that accompanies [14]. There are a number of small tuning details that affect performance in practice that we highlight. Our algorithm has essentially four hyperparameters: choice of $t$, splitting the privacy budget, radius of the clipping ball, and radius of the confidence ball. We explore the role of $t$ in our experiments. We found that assigning most of the privacy budget to the final iteration increased performance, namely $3\rho/4$ goes to the final iteration and $\rho/4(t-1)$ to each other. In theory, we use a relatively large value for the clipping threshold because it is more convenient for the analysis. In practice, we use a smaller clipping threshold to reduce sensitivity, which we find to improves practical performance.

**Experimental Setup.** We describe our setup for all the following experiments (later highlighting any relevant deviations). We generate a dataset of $n$ samples from a $d$-dimensional Gaussian $N(0, I)$, where we are promised the mean is bounded in $\ell_2$-distance by $R$. We run all the methods being compared to ensure a guarantee of $\rho$-CDP. We run each method 100 times, and report the trimmed mean, with trimming parameter $0.1$. We trim because a single failure can significantly inflate the average error. Our plots display the $\ell_2$-error of a method on the y-axis. We did not focus on optimizing the running time, but most of the plots (each involving about 1,000 runs of our algorithm) took only a few minutes to generate on a laptop computer with an Intel Core i7-7700HQ CPU.

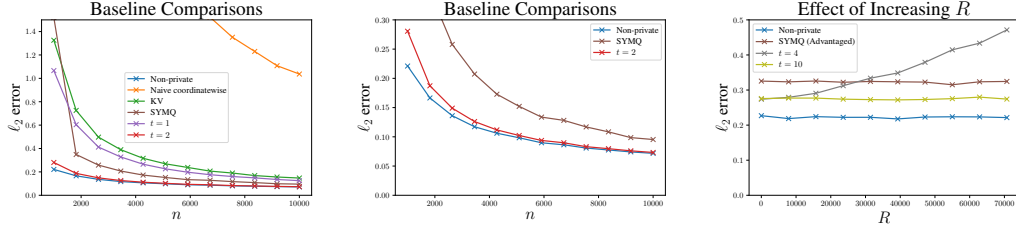

Figure 3: Baseline comparison of mean estimation algorithms (left and middle, $d = 50, R = 10\sqrt{d}, \rho = 0.5$). Our algorithm with $t = 2$ outperforms all other methods. Effect of increasing $R$ (right, $d = 50, \rho = 0.5, n = 1000$). Our method with $t = 10$ remains effective for very large $R$, with no perceptible disadvantage for small $R$.

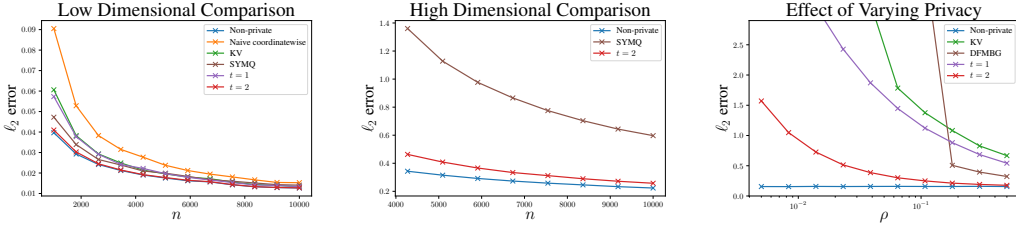

Figure 4: Comparison in low-dimensions $d = 2$ (left) and high-dimensions $d = 500$ (middle) with $\rho = 0.5$. Our method is superior in just 2 dimensions and is effective in 500 dimensions. Varying privacy level (right, $d = 50, n = 2000, R = 10\sqrt{d}$), our method is effective for small $\rho$.

**Results and Discussion.** In our first experiments (Figure 3, left and middle), we consider estimation in $d = 50$ dimensions with $\rho = 0.5$. We set the initial radius to $R = 10\sqrt{d}$, which roughly means that the user can estimate the mean of each coordinate a priori to within $\pm 10$ standard deviations. We then measure the error with varying choices of sample size $n$ between $10^3$ and $10^4$. The first and second panels show that our method (with $t = 2$ iterations) significantly outperforms previous methods, and offers error that is quite close to the non-private error. Concretely, we see that the additional cost of privacy is about 27% for $n = 1,000$ and decreases to just 2% for $n = 10,000$.

In our next experiment (Figure 3, right), we consider the effect of increasing the initial radius $R$, which corresponds to a user with less a priori knowledge of the parameters. Here, we fix $d = 50, n = 1,000, \rho = 0.5$ and vary $R$. We can see that when our method is run with $t = 10$ iterations, its error is essentially independent of $R$, showing no visible change in error as $R$ increases by three orders of magnitude. In contrast, all other methods show dramatically worse performance as $R$ grows. Note that, in these experiments, SYMQ is advantaged by receiving twice as many samples as our method—with the same number of samples SYMQ's error scaled linearly with $R$.

In the next set of experiments (Figure 4, left and middle) we consider both larger ($d = 500$) and smaller dimension ($d = 2$). In both cases we consider $R = 10\sqrt{d}$ and $\rho = 0.5$ and vary $n$. For bivariate data (left), our method (with $t = 2$ iterations) has the best performance, while other methods have error comparable to the naïve algorithm. For large dimension (middle), SYMQ is ineffective at small sample sizes while our method (with $t = 2$) competes well with non-private estimation—even with $n < 4d$ samples, the cost of privacy is less than a factor of 2.

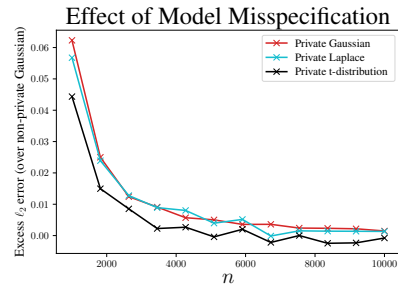

Figure 5: Mean estimation with non-Gaussian data ($d = 50, R = 10\sqrt{d}, \rho = 0.5$). Our method (with $t = 2$) is still effective for heavier-tailed data.

Next (Figure 4, right), we consider the effect of varying the privacy parameter $\rho$, fixing $d = 50, n = 1000, R = 10\sqrt{d}$. We observe that our methods remain superior for all choices of $\rho$, while coordinatewise methods are ineffective for high levels of privacy.

Lastly (Figure 5), we give evidence that our methods work for non-Gaussian distributions. We fix $d = 50, R = 10\sqrt{d}, \rho = 0.5$ and $t = 2$ iterations, and consider data drawn from various distributions

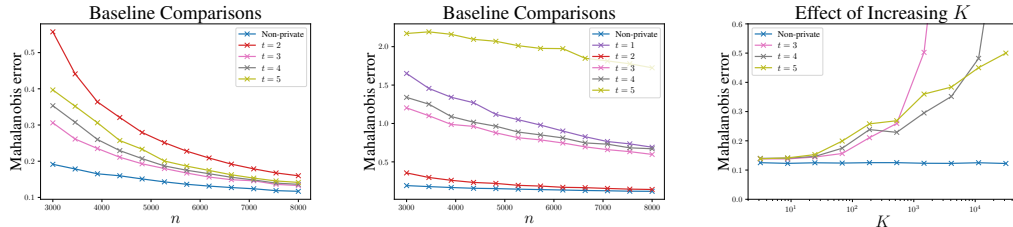

Figure 6: Baseline comparison of covariance estimation algorithms with isotropic (left) or skewed (middle) covariance, varying $n$ ($d = 10, K = 10\sqrt{d}, \rho = 0.5$). All values of $t > 1$ significantly outperform the $t = 1$ baseline, with the best choice of $t$ achieves a factor 1.5 more error than the non-private method at about $n = 3000$. Effect of incresing $K$ (right, $d = 10, n = 7000, \rho = 0.5$).

with heavier tails: the multivariate Laplace and Student's $t$-distribution with 3 degrees of freedom. We plot excess error compared to non-private estimation, observing that it is well behaved even for non-Gaussian data.

**Choice of $t$.** Our theory prescribes that the best choice of $t$ is $O(\log R)$. However, this is not extremely useful in practice, as it eliminates the constants which are key to choosing an appropriate value. Empirically, we found that choosing $t$ to be a small constant was effective, and the performance of the algorithm was relatively insensitive to generous settings of this parameter. In different experiments, the best performance uses different choices of $t$. However, we observe that in all experiments $t = 10$ performs competitively with the best choice of $t$ for that setting, showing that the method is relatively robust to how this hyperparameter is tuned. No other hyperparameters tuning was used between different experiments.

## 4 Covariance Estimation Experiments

We present our experimental results on multivariate covariance estimation. Additional experiments, as well as code to reproduce these experiments, appear in the supplement.

Covariance estimation offers fewer approaches to compare with, as it is unclear how to apply a univariate algorithm to the multivariate setting. In particular, estimating the off-diagonal terms of the covariance matrix is a very different problem than estimating the variance of a single normal, so an entrywise approach will run into significant challenges unless we assume the covariance matrix is diagonal. Thus, we compare the following two approaches: (1) Naïve method (MVCRᴇᴄ with $t = 1$)[7] (2) Our method (MVCRᴇᴄ for various $t > 1$). We do not compare with the algorithm of [24]. When implementing their algorithm, we found it too difficult to tune the numerous intertwined hyperparameters well enough to produce non-trivally accurate estimates. However, we note that our method can be seen as a "smoother" variant of their approach that is easier to implement and tune.

**Implementation Details.** Full details of the algorithm appear in our accompanying code. We find that the same optimizations to the choice of $t$ and the division of the privacy budget $\rho$ are helpful for covariance estimation. However, for covariance estimation we also find that an even more aggressive shrinking of the confidence ellipsoids gives the best concrete performance.

### 4.1 Synthetic Data Experiments

**Experimental Setup.** We describe our setup for the following experiments, highlighting any relevant deviations later. We generate a dataset of $n$ samples from a $d$-dimensional Gaussian, either $N(0, I)$ (the isotropic case), or $N(0, \Sigma)$, where $\Sigma$ is a random rotation of a matrix with $d/2$ eigenvalues equal to each of the values $K$ and 1 (the skewed case). As with mean estimation, we run 100 trials of the algorithm and report the trimmed mean with trimming level 0.1. Our plots display the Mahalanobis error of a method on the y-axis, or, equivalently, the Frobenius error after accounting for differences in scaling in all directions. All experiments were completed within a few minutes on a laptop computer with an Intel Core i7-7700HQ CPU.

**Results and Discussion.** In our first experiments (Figure 6, left and middle), we vary $n$ while keeping other parameters fixed ($d = 10, K = 10\sqrt{d}, \rho = 0.5$). Note that with $d = 10$ dimensions, the covariance matrix has $\binom{d+1}{2} = 55$ non-redundant parameters, so these experiments are somewhat comparable to our experiments on mean estimation with $d = 50$ parameters. We first examine the isotropic case (left) where our method (with any $t > 1$) significantly outperforms the baseline method ($t = 1$). For $t = 3$ we are competitive within a factor of

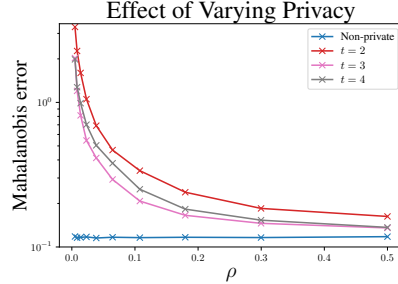

Figure 7: Varying the privacy level ($d = 10, n = 8000, K = 10\sqrt{d}$) with isotropic covariance.

1.5 for $n > 3000$. For the skewed case (middle) we obtain similar competitiveness, albeit using the $t = 2$ variant of our algorithm instead. The fact that the algorithm's performance is sensitive to the choice of $t$, which was not true for mean estimation, is an interesting phenomenon for further study, although private hyperparameter tuning of this sort can be addressed using [28].

Next, (Figure 6, right), we fix other parameters ($d = 10, n = 7000, \rho = 0.5$, isotropic covariance) consider the effect of increasing the initial radius $K$, simulating a user with less initial knowledge of the covariance. We see that the method performs well, with error growing roughly logarithmically in $K$, although larger choices of $t$ appear optimal when $K$ is large.

Lastly (Figure 7), we vary the privacy parameter $\rho$, fixing other parameters ($d = 50, n = 8000, K = 10\sqrt{d}$), showing that our method performs well as the privacy level increases.

## 4.2 Map of Europe

To demonstrate a practical use of our algorithm, we investigate an application of our method to private principal component analysis, a core technique used by data scientists in exploratory data analysis. We revisit the classic "genes mirror geography" discovery of Novembre et al. [30]. In this work, the authors investigated a multivariate dataset collected as part of the Population Reference Sample (POPRES) project. This dataset contained the genetic data of over 1387 European individuals, annotated by their country of origin. The authors projected this dataset onto its top two principal components to produce a two-dimensional representation of the genetic variation, which bears a strong resemblance to the map of Europe. However, a significant pitfall is that the given dataset is highly sensitive in nature, consisting of individuals' genetic data. As such, it would be advantageous if we could extract the same insights from the data, even under the constraint of differential privacy. To this end, we investigate the efficacy of our method in comparison to the baseline private method.

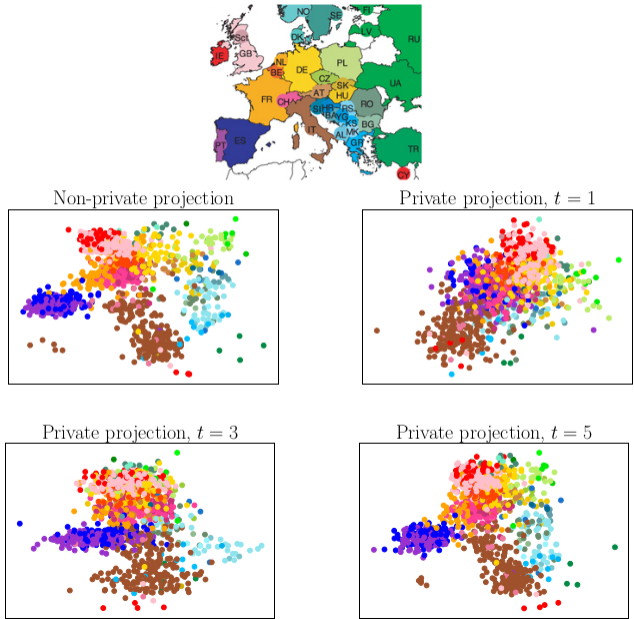

Figure 8: Privately recovering Europe. Top: color-coded map of Europe, middle-left: non-private projection, middle-right: naïve private baseline ($t = 1$). Most structure is lost. Our results (bottom, $t = 3$ and $t = 5$) are much more effective at privately estimating the projection.

Though the original dataset is very high dimensional, we obtained a 20-dimensional version of the dataset ($n = 1387$) from the authors' GitHub.[8] We randomly rotate the data to ensure that any structure in their representation is lost. Recall that the analyst must have some prior knowledge about the covariance: in our algorithm's phrasing, they must select a parameter $K$ and a transformation of the data which places the true

[8] https://github.com/NovembreLab/Novembre_etal_2008_misc

covariance between $I$ and $KI$. We simulate an analyst who has minimal information about the data, scaling up the data by a factor of 20 and setting $K = 30$ – the true top two eigenvalues afterwards are roughly 4.8 and 1.2, so the loose upper bound on $K$ signifies that there is significant uncertainty on the scale of the data. An analyst with even less information could pick a larger scaling factor and $K$.

Our results are presented in Figure 8. The first subplot shows the results of the experiment using the non-private empirical covariance. The second subplot is the private projection of the dataset using the naïve method ($t = 1$). We can see that most of the structure is lost – the inner product of the top two private principal components with the true ones are 0.48 and 0.28. The third subplot is the private projection of the dataset when we use our method with $t = 3$. This bears a stronger resemblance to the original image – the same dot products are now 0.98 and 0.48. As the top principal component is much larger in magnitude than the second one, it is easier to accurately recover. Finally, the fourth subplot is the private projection of our method with $t = 5$. This bears the strongest resemblance to the original image, with dot products of 0.96 and 0.92. Thus, our method demonstrates promise for improving performance of private exploratory data analysis.

## 5    Conclusions

We gave the first practical algorithms for differentially private estimation of mean and covariance in the multivariate setting. We showed that these algorithms have strong theoretical guarantees (matching the state-of-the-art), and are accurate even at relatively low sample sizes and high dimensions. They significantly outperform all prior methods, possess few hyperparameters, and remain precise when given minimal prior info about the data. We also showed that our methods can be used for private PCA, a task which is common in data science and exploratory data analysis. As we are seeing the rise of a number of new libraries for practical differentially private statistics and data analysis [31, 1] we believe our results add an important tool to the toolkit for the multivariate setting.

## Broader Impact

Our work provides realizable tools for private data analysis. Given recent concerns centered around large-scale data collection and surveillance, the production of a mature and robust set of tools which preserve privacy can help assuage public fears involving misuse of personal data. Additionally, we hope that developing tools which approach the non-private accuracy will inspire companies to adopt privacy by default.

As differential privacy requires technical domain knowledge, incorrect use or misinterpretation of differential privacy is unfortunately easy and can lead to negative side effects including providing a misleading or false sense of security. Such issues can be avoided by sufficient training and/or consultation with experts in data privacy, although this may present more of a challenge for smaller, resource-constrained organizations.

## Acknowledgments and Disclosure of Funding

SB was supported by a University of Waterloo startup grant. GK was supported by an NSERC Discovery grant, a Compute Canada RRG grant, and a University of Waterloo startup grant. JU was supported by NSF grants CCF-1718088, CCF-1750640, CNS-1816028, and CNS-1916020

GK would like to thank Aleksandar Nikolov for useful discussions about the proof of Lemma 2.3.

## Footnotes

*Authors ordered alphabetically. Full version of the paper and code are available in the supplement, or alternatively at the following URLs: https://arxiv.org/abs/2006.06618 and https://github.com/twistedcubic/coin-press.

[2]While our results are stated for Gaussian data, we comment that identical arguments hold for sub-Gaussian distributions as well.

[3]Under pure or concentrated DP, the dependence on $R$ and $K$ must be polylogarithmic [27, 5]. One can allow $R = \infty$ for mean estimation under $(\varepsilon, \delta)$-DP with $\delta > 0$ [27], although the resulting algorithm has poor concrete performance even for univariate data. It is an open question whether one can allow $K = \infty$ for covariance estimation even under $(\varepsilon, \delta)$-DP.

[4]For simplicity, we use the common convention that the size of the sample $n$ is fixed and public.

[5]We assume familiarity with basic properties of differential privacy, which appear in the supplement.

[6]zCDP and DP are on different scales, but otherwise can be ordered from most-to-least restrictive. Specifically, $(\varepsilon, 0)$-DP implies $\frac{\varepsilon^2}{2}$-zCDP, which implies $(\varepsilon\sqrt{\log(1/\delta)}, \delta)$-DP for every $\delta > 0$ [6].

[7]This method, which amounts to clipping the data and then adding noise to the empirical covariance, is sometimes called "Analyze Gauss," due to its use in [19].

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
