[Supplementary Material]

# CoinPress: Practical Private Mean and Covariance Estimation[*]

Sourav Biswas[†]    Yihe Dong[‡]    Gautam Kamath[§]    Jonathan Ullman[¶]

October 18, 2020

## Abstract

> We present simple differentially private estimators for the mean and covariance of multivariate sub-Gaussian data that are accurate at small sample sizes. We demonstrate the effectiveness of our algorithms both theoretically and empirically using synthetic and real-world datasets—showing that their asymptotic error rates match the state-of-the-art theoretical bounds, and that they concretely outperform all previous methods. Specifically, previous estimators either have weak empirical accuracy at small sample sizes, perform poorly for multivariate data, or require the user to provide strong a priori estimates for the parameters.

## 1 Introduction

One of the most basic problems in statistics and machine learning is to estimate the mean and covariance of a distribution based on i.i.d. samples. Not only are these some of the most basic summary statistics one could want for a real-valued distribution, but they are also building blocks for more sophisticated statistical estimation tasks like linear regression and stochastic convex optimization.

The optimal solutions to these problems are folklore—simply output the empirical mean and covariance of the samples. However, this solution is not suitable when the samples consist of sensitive, private information belonging to individuals, as it has been shown repeatedly that even releasing just the empirical mean can reveal this sensitive information [DN03, HSR+08, BUV14, DSS+15, DSSU17]. Thus, we need estimators that are not only accurate with respect to the underlying distribution, but also protect the *privacy* of the individuals represented in the sample.

The most widely accepted solution to individual privacy in statistics and machine learning is *differential privacy* (DP) [DMNS06], which provides a strong guarantee of individual privacy by ensuring that no individual has a significant influence on the learned parameters. A large body of work now shows that, in principle, nearly every statistical task can be solved privately, and differential privacy is now being deployed by Apple [Dif17], Google [EPK14, BEM+17], Microsoft [DKY17], and the US Census Bureau [DLS+17].

---

[*]Authors ordered alphabetically. Code is available at `https://github.com/twistedcubic/coin-press`.

[†]Cheriton School of Computer Science, University of Waterloo. `s23biswa@uwaterloo.ca`. Supported by a University of Waterloo startup grant.

[‡]Microsoft. `yihdong@microsoft.com`.

[§]Cheriton School of Computer Science, University of Waterloo. `g@csail.mit.edu`. Supported by a University of Waterloo startup grant and a Compute Canada RRG grant.

[¶]Khoury College of Computer Sciences, Northeastern University. `jullman@ccs.neu.edu`. Supported by NSF grants CCF-1718088, CCF-1750640, CNS-1816028, and CNS-1916020.

Figure 1: The cost of privacy, measured as the ratio of our iterative estimator's error to that of the non-private estimator. For mean estimation (left) we use $d = 50$ and privacy level $\rho = 0.5$ and vary $n \in (300, 5000)$. For covariance estimation (right) we use $d = 10$ privacy level $\rho = 0.5$ and vary $n \in (2000, 10000)$.

Differential privacy requires adding random noise to some stage of the estimation procedure, and this noise might increase the error of the final estimate. Typically, the amount of noise vanishes as the sample size $n$ grows, and one can often show that as $n \to \infty$, the additional error due to privacy vanishes faster than the sampling error of the estimator, making differential privacy highly practical for large samples.

However, differential privacy is often difficult to achieve for small datasets, or when the dataset is large, but we want to restrict attention to some small subpopulation within the data. Thus, a recent trend has been to focus on simple, widely used estimation tasks, and design estimators with good concrete performance at small samples sizes. Most relevant to our work, Karwa and Vadhan [KV18] and Du, Foot, Moniot, Bray, and Groce [DFM+20] give practical mean and variance estimators for univariate Gaussian data. However, as we show, these methods do not scale well to the more challenging multivariate setting.

## 1.1 Contributions

In this work we give simple, practical estimators for the mean and covariance of *multivariate* sub-Gaussian data. We call our method COINPRESS, for COnfidence-INterval-based PRivate EStimation Strategy. We validate our estimators theoretically and empirically. On the theoretical side, we show that our estimators match the state-of-the-art asymptotic bounds for sub-Gaussian mean and covariance estimation [KLSU19]. On the empirical side, we give an extensive evaluation with synthetic data, as well as a demonstration on a real-world dataset. We show that our estimators have error comparable to that of the non-private empirical mean and covariance at small sample sizes. See Figure 1 for one representative example of our algorithm's performance. Our mean estimator also improves over the state-of-the-art method of Du et al. [DFM+20], which was developed for univariate data but can be applied coordinate-wise to estimate multivariate data. We highlight a few other important features of our methods:

First, like many differentially private estimators, our method requires the user to input some a priori knowledge of the data. For mean estimation, we require the mean lives in a specified ball of radius $R$, and for covariance estimation we require that the covariance matrix can be sandwiched spectrally between $A$ and $KA$ for some matrix $A$. Some a priori boundedness is

*necessary* for algorithms like ours that satisfy concentrated DP [DR16, BS16, Mir17, BDRS18], or satisfy pure DP.[1] We show that our estimator is practical when these parameters are taken to be extremely large, meaning the user only needs a very weak prior.

Second, for simplicity, we describe and evaluate our methods primarily with Gaussian data. However, the only feature of Gaussian data that is relevant for our methods is a strong bound on the tails of the distribution, and, by definition, these bounds hold for any sub-Gaussian distribution. Moreover, using experiments with both heavier-tailed synthetic data and with real-world data, we demonstrate that our method remains useful even when the data is not truly Gaussian. Note that some restriction on the details of the data is necessary, at least in the worst-case, as [KSU20] showed that the minimax optimal error is highly sensitive to the rate of decay of the distribution's tails.

**Approach.** At a high-level, our estimators work by iteratively refining an estimate for the parameters, inspired by [KLSU19]. For mean estimation, we start with some (potentially very large) ball $B_1$ of radius $R_1$ that we know contains most of the mass of the probability distribution. We then use this ball to run a naïve estimation procedure: clip the data to the ball $B_1$, then add noise to the empirical mean of the clipped data to obtain some initial estimate of the mean. Specifically, the noise will have magnitude proportional to $R_1/n$. Using this estimate, and knowledge of how we obtainesd it, we can draw a (hopefully significantly smaller) ball $B_2$ of radius $R_2$ that contains most of the mass and then repeat. After a few iterations, we will have some ball $B_t$ of radius $R_t$ that tightly contains most of the datapoints, and use this to make an accurate final private estimate of the meanwith noise proportional to $R_t/n$. Our covariance estimation uses the same iterative approach, although the geometry is significantly more subtle.

Figure 2: Visualizing a run of the mean estimator with $n = 160, \rho = 0.1, t = 3$. The data is represented by the blue dots, the black circles represent the iteratively shrinking confidence ball, and the orange dot is the final private mean estimate.

## 1.2 Problem Formulation

We now give a more detailed description of the problem we consider in this work. We are given an ordered set of samples $X = (X_1, \ldots, X_n) \subseteq \mathbb{R}^d$ where each $X_i$ represents some individual's sensitive data. We would like an estimator $M(X)$ that is *private* for the individuals in the sample, and also *accurate* in that when $X$ consists of i.i.d. samples from some distribution $P$, then $M(X)$ estimates the mean and covariance of $P$ with small error. Notice that privacy will be a *worst-case* property, making no distributional assumptions, whereas accuracy will be formulated as an *average-case* property relying on distributional assumptions.

For privacy, we require that $M$ is insensitive to any one sample in $X$ in the following sense: We say that two samples $X, X' \subseteq \mathbb{R}^d$ of size $n$ are *neighboring* if they differ on at most one

sample.[2] Informally, we say that a randomized algorithm $M$ is *differentially private* [DMNS06] if the distributions $M(X)$ and $M(X')$ are similar for every pair of neighboring datasets $X, X'$. In this work, we adopt the quantiative formulation of differential privacy called *concentrated differential privacy (zCDP)* [DR16, BS16].

**Definition 1.1** (zCDP). *An estimator $M(X)$ satisfies $\rho$-zCDP if for every pair of neighboring samples $X, X'$ of size $n$, and every $\alpha \in (1, \infty)$, $D_\alpha(M(X)\|M(X')) \leq \rho\alpha$, where $D_\alpha$ is the Rényi divergence of order $\alpha$.*

This formulation sits in between general $(\varepsilon, \delta)$-differential privacy and the special case of $(\varepsilon, 0)$-differential privacy,[3] and better captures the privacy cost of private algorithms in high dimension [DSSU17].

To formulate the accuracy of our mechanism, we posit that $X$ is sampled i.i.d. from some distribution $P$, and our goal is to estimate the mean $\mu \in \mathbb{R}^d$ and covariance $\Sigma \in \mathbb{R}^{d \times d}$ with

$$\mu = \mathbb{E}_{x \sim P}[x] \quad \text{and} \quad \Sigma = \mathbb{E}_{x \sim P}\big[(x - \mu)^T(x - \mu)\big].$$

We assume that our algorithms are given some a priori estimate of the mean in the form of a radius $R$ such that $\|\mu\|_2 \leq R$ and some a priori estimate of the covariance in the form of $K$ such that $I \preceq \Sigma \preceq KI$, (equivalently all the singular values of $\Sigma$ lie between 1 and $K$). We repeat that some a priori bound on $R, K$ is *necessary* for any algorithm that satisfies zCDP [BS16, KV18].

We measure the error in *Mahalanobis* distance $\|\cdot\|_\Sigma$, which reports how the error compares to the covariance of the distribution, and has the benefit of being invariant under affine transformations. Specifically,

$$\|\hat\mu - \mu\|_\Sigma = \|\Sigma^{-1/2}(\hat\mu - \mu)\|_2 \quad \text{and} \quad \|\hat\Sigma - \Sigma\|_\Sigma = \|\Sigma^{-1/2}\hat\Sigma\Sigma^{-1/2} - I\|_F.$$

For any distribution, the *empirical mean* $\hat\mu$ and *empirical covariance* $\hat\Sigma$ satisfy

$$\mathbb{E}[\|\hat\mu - \mu\|_\Sigma] \leq \sqrt{\frac{d}{n}} \quad \text{and} \quad \mathbb{E}\Big[\|\hat\Sigma - \Sigma\|_\Sigma\Big] \leq \sqrt{\frac{d^2}{n}},$$

and these estimators are minimax optimal. Our goal is to obtain estimators that have similar accuracy to the empirical mean and covariance. We note that the folklore naïve estimators (see e.g. [KV18, KLSU19]) for mean and covariance based on clipping the data to an appropriate ball and adding carefully calibrated noise to the empirical mean and covariance would guarantee

$$\mathbb{E}[\|\hat\mu - \mu\|_\Sigma] \leq \sqrt{\frac{d}{n} + \frac{R^2 d^2}{n^2\rho}} \quad \text{and} \quad \mathbb{E}\Big[\|\hat\Sigma - \Sigma\|_\Sigma\Big] \leq \sqrt{\frac{d^2}{n} + \frac{K^2 d^4}{n^2\rho}}.$$

The main downside of the naïve estimators is their error increases rapidly with $R$ and $K$, and thus introduce large error unless the user has strong a priori knowledge of the mean and covariance. Requiring users to provide such a priori bounds is a major challenge in deployed systems for differentially private analysis (e.g. [GHK+16]). Our estimators have much better dependence on these parameters, both asymptotically and concretely.

For our theoretical analysis and most of our evaluation, we derive bounds on the error of our estimators assuming $P$ is specifically the Gaussian $N(\mu, \Sigma)$. Although, as we show in some of our experiments, our methods perform well even when we relax this assumption.

## 1.3 Related Work

The most relevant line of work is that initiated by Karwa and Vadhan [KV18], which studies private mean and variance estimation for Gaussian data, and focuses on important issues for practice like dealing with weak a priori bounds on the parameters. Later works studied the multivariate setting [KLSU19, CWZ19, KSSU19] and estimation under weaker moment assumptions [BS19, KSU20], though these investigations are primarily theoretical. Our algorithm for covariance estimation can be seen as a simpler and more practical variant of [KLSU19]. They provide an iterative procedure which, based on a privatized finds the subspaces of high and low variance they iteratively threshold eigenvalues to find directions of high and low variance, whereas we employ a softer method to avoid wasting information. One noteworthy work is [DFM+20], which provides practical private confidence intervals in the univariate setting. Instead, our investigation is focused on realizable algorithms for the multivariate setting. Several works consider private PCA or covariance estimation [DTTZ14, HP14, ADK+19], though, unlike our work, these methods assume strong a priori bounds on the covariance.

A number of the early works in differential privacy give methods for differentially private statistical estimation for i.i.d. data. The earliest works [DN03, DN04, BDMN05, DMNS06], which introduced the Gaussian mechanism, among other foundational results, can be thought of as methods for estimating the mean of a distribution over the hypercube $\{0,1\}^d$ in the $\ell_\infty$ norm. Tight lower bounds for this problem follow from the tracing attacks introduced in [BUV14, SU17a, DSS+15, BSU17, SU17b]. A very recent work of Acharya, Sun, and Zhang [ASZ20] adapts classical tools for proving estimation and testing lower bounds (the lemmata of Assouad, Fano, and Le Cam) to the private setting. Steinke and Ullman [SU17b] give tight minimax lower bounds for the weaker guarantee of selecting the largest coordinates of the mean, which were refined by Cai, Wang, and Zhang [CWZ19] to give lower bounds for sparse mean-estimation.

Other approaches for Gaussian estimation include [NRS07], which introduced the sample-and-aggregate paradigm, and [BKSW19] which employs a private hypothesis selection method. Zhang, Kamath, Kulkarni, and Wu privately estimate Markov Random Fields [ZKKW20], a generalization of product distributions over the hypercube. Dwork and Lei [DL09] introduced the propose-test-release framework for estimating robust statistics such as the median and interquartile range. For further coverage of private statistics, see [KU20].

## 2 Preliminaries

We begin by recalling the definition of differential privacy, and the variant of concentrated differential privacy that we use in this work.

**Definition 2.1** (Differential Privacy (DP) [DMNS06]). *A randomized algorithm $M : \mathcal{X}^n \to \mathcal{Y}$ satisfies $(\varepsilon, \delta)$-differential privacy ($(\varepsilon, \delta)$-DP) if for every pair of neighboring datasets $X, X' \in \mathcal{X}^n$ (i.e., datasets that differ in exactly one entry),*

$$\forall Y \subseteq \mathcal{Y} \quad \mathbb{P}[M(X) \in Y] \leq e^\varepsilon \cdot \mathbb{P}[M(X') \in Y] + \delta.$$

*When $\delta = 0$, we say that $M$ satisfies $\varepsilon$-differential privacy or pure differential privacy.*

**Definition 2.2** (Concentrated Differential Privacy (zCDP) [BS16]). *A randomized algorithm $M : \mathcal{X}^n \to \mathcal{Y}$ satisfies $\rho$-zCDP if for every pair of neighboring datasets $X, X' \in \mathcal{X}^n$,*

$$\forall \alpha \in (1, \infty) \quad D_\alpha\big(M(X)||M(X')\big) \leq \rho\alpha,$$

where $D_\alpha(M(X)\|M(X'))$ is the $\alpha$-Rényi divergence between $M(X)$ and $M(X')$.[4]

Note that zCDP and DP are on different scales, but are otherwise can be ordered from most-to-least restrictive. Specifically, $(\varepsilon, 0)$-DP implies $\frac{\varepsilon^2}{2}$-zCDP, which implies roughly $(\varepsilon\sqrt{2\log(1/\delta)}, \delta)$-DP for every $\delta > 0$ [BS16].

Both these definitions are closed under post-processing and can be composed with graceful degradation of the privacy parameters.

**Lemma 2.3** (Post Processing [DMNS06, BS16]). *If $M : \mathcal{X}^n \to \mathcal{Y}$ is $(\varepsilon, \delta)$-DP, and $P : \mathcal{Y} \to \mathcal{Z}$ is any randomized function, then the algorithm $P \circ M$ is $(\varepsilon, \delta)$-DP. Similarly if $M$ is $\rho$-zCDP then the algorithm $P \circ M$ is $\rho$-zCDP.*

**Lemma 2.4** (Composition of CDP [DMNS06, DRV10, BS16]). *If $M$ is an adaptive composition of differentially private algorithms $M_1, \ldots, M_T$, then*

1. *if $M_1, \ldots, M_T$ are $(\varepsilon_1, \delta_1), \ldots, (\varepsilon_T, \delta_T)$-DP then $M$ is $(\sum_t \varepsilon_t, \sum_t \delta_t)$-DP, and*

2. *if $M_1, \ldots, M_T$ are $\rho_1, \ldots, \rho_T$-zCDP then $M$ is $(\sum_t \rho_t)$-zCDP.*

We can achieve differential privacy via noise addition proportional to sensitivity [DMNS06].

**Definition 2.5** (Sensitivity). *Let $f : \mathcal{X}^n \to \mathbb{R}^d$ be a function, its $\ell_2$-sensitivity is defined to be $\Delta_{f,2} = \max_{X \sim X' \in \mathcal{X}^n} \|f(X) - f(X')\|_2$. Here, $X \sim X'$ denotes that $X$ and $X'$ are neighboring datasets (i.e., those that differ in exactly one entry).*

For functions with bounded $\ell_1$-sensitivity, we can achieve $\varepsilon$-DP by adding noise from a Laplace distribution proportional to $\ell_1$-sensitivity. For functions taking values in $\mathbb{R}^d$ for large $d$ it is more useful to add noise from a Gaussian distribution proportional to the $\ell_2$-sensitivity, to get $(\varepsilon, \delta)$-DP and $\rho$-zCDP.

**Lemma 2.6** (Gaussian Mechanism). *Let $f : \mathcal{X}^n \to \mathbb{R}^d$ be a function with $\ell_2$-sensitivity $\Delta_{f,2}$. Then the Gaussian mechanism*

$$M_f(X) = f(X) + N\left(0, \left(\frac{\Delta_{f,2}}{\sqrt{2\rho}}\right)^2 \cdot I_{d \times d}\right)$$

*satisfies $\rho$-zCDP.*

# 3  New Algorithms for Multivariate Gaussian Estimation

In this section, we present new algorithms for Gaussian parameter estimation. While these do not result in improved asymptotic sample complexity bounds, they will lead to algorithms which are much more practical in the multivariate setting. In particular, they will avoid the curse of dimensionality incurred by multivariate histograms, but also eschew many of the hyperparameters that arise in previous methods [KLSU19]. Note that our algorithms with $t = 1$ precisely describe the naïve method that was informally outlined in Section 1.2. We describe the algorithms and sketch ideas behind the proofs, which appear in the appendix. Also in the appendix, we describe simpler univariate algorithms in the same style. Understanding these algorithms and proofs first might be helpful before approaching algorithms for the multivariate setting.

**Algorithm 1** One Step Private Improvement of Mean Ball

---

    **Input:** $n$ samples $X_{1...n}$ from $N(\mu, I_{d \times d})$, $B_2(c, r)$ containing $\mu$, $\rho_s, \beta_s > 0$

    **Output:** A $\rho_s$-zCDP ball $B_2(c', r')$

1:  **procedure** MVM$(X_{1...n}, c, r, \rho_s, \beta_s)$

2:     Let $\gamma_1 = \sqrt{d + 2\sqrt{d \log(n/\beta_s)} + 2\log(n/\beta_s)}$.

3:     Let $\gamma_2 = \sqrt{d + 2\sqrt{d \log(1/\beta_s)} + 2\log(1/\beta_s)}$.                    $\triangleright \gamma_1, \gamma_2 \approx \sqrt{d}$

4:     Project each $X_i$ into $B_2(c, r + \gamma_1)$.

5:     Let $\Delta = 2(r + \gamma_1)/n$.

6:     Compute $Z = \frac{1}{n}\sum_i X_i + Y$, where

$$Y \sim N\left(0, \left(\frac{\Delta}{\sqrt{2\rho_s}}\right)^2 \cdot I_{d \times d}\right).$$

7:     Let $c' = Z$, $r' = \gamma_2 \sqrt{\frac{1}{n} + \frac{2(r+\gamma_1)^2}{n^2 \rho_s}}$.

8:     **return** $(c', r')$.

9: **end procedure**

---

---

**Algorithm 2** Private Confidence-Ball-Based Multivariate Mean Estimation

---

    **Input:** $n$ samples $X_{1...n}$ from $N(\mu, I_{d \times d})$, $B_2(c, r)$ containing $\mu$, $t \in \mathbb{N}^+$, $\rho_{1...t}, \beta > 0$

    **Output:** A $(\sum_{i=1}^{t} \rho_i)$-zCDP estimate of $\mu$

1:  **procedure** MVMREC$(X_{1...n}, c, r, t, \rho_1, \ldots, \rho_t, \beta)$

2:     Let $(c_0, r_0) = (c, r)$.

3:     **for** $i \in [t-1]$ **do**

4:         $(c_i, r_i) = \text{MVM}(X_{1...n}, c_{i-1}, r_{i-1}, \rho_i, \frac{\beta}{4(t-1)})$.

5:     **end for**

6:     $(c_t, r_t) = \text{MVM}(X_{1...n}, c_{t-1}, r_{t-1}, \rho_t, \frac{\beta}{4})$.

7:     **return** $c_t$.

8: **end procedure**

---

### 3.1   Multivariate Private Mean Estimation

We first present our multivariate private mean estimation algorithm MVMREC (Algorithm 2). This is an iterative algorithm, which maintains a confidence ball that contains the true mean with high probability. For ease of presentation, we state the algorithm for a Gaussian with identity variance. However, by rescaling the data, the same argument works for an arbitrary known covariance $\Sigma$. In fact, the covariance doesn't even need to be known exactly – we just need a proxy $\hat{\Sigma}$ such that $C_1 I \preceq \hat{\Sigma}^{-1/2} \Sigma \hat{\Sigma}^{-1/2} \preceq C_2 I$, where $0 < C_1 < C_2$ are absolute constants.

    MVMREC calls MVM $t - 1$ times, each time with a new $\ell_2$-ball $B_2(c_i, r_i)$ centered at $c_i$ with radius $r_i$. We desire that each invocation is such that $\mu \in B_2(c_i, r_i)$, and the $r_i$'s should decrease rapidly, so that we quickly converge to a fairly small ball which contains the mean. Our goal will be to acquire a small enough radius. With this in hand, we can run the naïve algorithm which clips the data and applies the Gaussian mechanism. With large enough $n$, this will have the desired accuracy.

    It remains to reason about MVM. We need to argue (a) privacy and (b) accuracy: given

$B_2(c, r) \ni \mu$, it is likely to output $B_2(c', r') \ni \mu$; and c) progress: the radius $r'$ output is much smaller than the $r$ input. The algorithm first chooses some $\gamma$ and clips the data to $B_2(c, r + \gamma)$. $\gamma$ is chosen based on Gaussian tail bounds such that if $\mu \in B_2(c, r)$, then none of the points will be affected by this operation. This bounds the sensitivity of the empirical mean, and applying the Gaussian mechanism guarantees privacy. While the noised mean will serve as a point estimate $c'$, we actually have more: again using Gaussian tail bounds on the data combined with the added noise, we can define a radius $r'$ such that $\mu \in B_2(c', r')$, establishing accuracy. Finally, a large enough $n$ will ensure $r' < r/2$, establishing progress. Since each step reduces our radius $r_i$ by a constant factor, setting $t = O(\log R)$ will reduce the initial radius from $R$ to $O(1)$ as desired. Formalizing this gives the following theorem.

**Theorem 3.1.** MVMREC *is* $(\sum_{i=1}^{t} \rho_i)$-*zCDP. Furthermore, suppose* $X_1, \ldots, X_n$ *are samples from* $N(\mu, I)$ *with* $\mu$ *contained in the ball* $B_2(C, R)$, *and* $n = \tilde{\Omega}((\frac{d}{\alpha^2} + \frac{d}{\alpha\sqrt{\rho}} + \frac{\sqrt{d \log R}}{\sqrt{\rho}}) \cdot \log \frac{1}{\beta})$. *Then* MVMREC$(X_1, \ldots, X_n, C, R, t = O(\log R), \frac{\rho}{2(t-1)}, \ldots, \frac{\rho}{2(t-1)}, \frac{\rho}{2}, \beta)$ *will return* $\hat{\mu}$ *such that* $\|\mu - \hat{\mu}\|_\Sigma = \|\mu - \hat{\mu}\|_2 \leq \alpha$ *with probability at least* $1 - \beta$.

## 3.2 Multivariate Private Covariance Estimation

We describe our multivariate private covariance estimation algorithm MVCREC (Algorithm 4). The ideas are conceptually similar to mean estimation, but subtler due to the more complex geometry. We assume data is drawn from $N(0, \Sigma)$ where $I \preceq \Sigma \preceq KI$ for some known $K$. One can reduce to the zero-mean case by differencing pairs of samples. Further, if we know some PSD matrix $A$ such that $A \preceq \Sigma \preceq KA$ then we can rescale the data by $A^{-1/2}$.

To specify the algorithm we need a couple of tail bounds on the norm of points from a normal distribution, the spectral-error of the empirical covariance, and the spectrum of a certain random matrix. As these expressions are somewhat ugly, we define them outside of the pseudocode. In these expressions, we fix parameters $n, d, \beta_s$.

$$\gamma = \sqrt{d + 2\sqrt{d \log(n/\beta_s)} + 2 \log(n/\beta_s)}$$

$$\eta = 2\left(\sqrt{\frac{d}{n}} + \sqrt{\frac{2 \ln(\beta_s/2)}{n}}\right) + \left(\sqrt{\frac{d}{n}} + \sqrt{\frac{2 \ln(\beta_s/2)}{n}}\right)^2 \tag{1}$$

$$\nu = \left(\frac{\gamma^2}{n\sqrt{\rho_s}}\right)\left(2\sqrt{d} + 2d^{1/6} \log^{1/3} d + \frac{6(1 + ((\log d)/d)^{1/3})\sqrt{\log d}}{\sqrt{\log(1 + (\log d/d)^{1/3})}} + 2\sqrt{2\log(1/\beta_s)}\right) \tag{2}$$

Similar to MVMREC, MVCREC repeatedly calls a private algorithm (MVC) that makes a constant-factor progress, and then run the naïve algorithm. Rather than maintaining a ball containing the true mean, we maintain a pair of ellipsoids which sandwich the true covariance between them (in fact, we rescale at each step so that the outer covariance is always the identity).[5] Once the axes of these ellipsoids are within a constant factor we can apply the naïve algorithm, which is accurate given enough samples.

The algorithm MVC is similar to MVM. We first clip the points at a distance based on Gaussian tail bounds with respect to the outer ellipsoid, which is unlikely to affect the dataset when it dominates the true covariance. After this operation, we can show that the sensitivity

**Algorithm 3** One Step Private Improvement of Covariance Ball

---
    **Input:** $n$ samples $X_{1\ldots n}$ from $N(0,\Sigma)$, matrices $A, L$ such that $L \preceq A\Sigma A \preceq I$, $\rho_s, \beta_s > 0$
    **Output:** A $\rho_s$-zCDP lower bound matrix $L'$, symmetric matrix $A'$, noised covariance $Z$

1:  **procedure** $\text{MVC}(X_{1\ldots n}, A, L, \rho_s, \beta_s)$
2:     Compute $W_i = AX_i$.                             $\triangleright\ W_i \sim N(0, A\Sigma A), L \preceq A\Sigma A \preceq I$
3:     Let $\gamma = \sqrt{d + 2\sqrt{d\log(n/\beta_s)} + 2\log(n/\beta_s)}$.                   $\triangleright\ \gamma \approx \sqrt{d}$
4:     Project each $W_i$ int $B_2(0, \gamma)$.
5:     Let $\Delta = \sqrt{2}\gamma^2/n$.
6:     Compute $Z = \frac{1}{n}\sum_i W_i W_i^T + Y$, where $Y$ is the $d \times d$ matrix with independent $N(0, \Delta^2/2\rho_s^2)$ entries in the upper triangle and diagonal, and then made symmetric.
7:     Let $\eta, \nu$ be as defined in (1) and (2), respectively.        $\triangleright\ \eta \approx \sqrt{\frac{d}{n}}$ and $\nu \approx \frac{\gamma^2\sqrt{d}}{n\sqrt{\rho_s}}$
8:     Let $\tilde{U} = Z + \eta I$.
9:     Let $L' = \tilde{U}^{-1/2}(Z - \eta I - \nu I)\tilde{U}^{-1/2}$ and $A' = \tilde{U}^{-1/2}A$.
10:    **return** $L', A', Z$.
11: **end procedure**

---

---

**Algorithm 4** Private Confidence-Ball-Based Multivariate Covariance Estimation

---
    **Input:** $n$ samples $X_{1\ldots n}$ from $N(0,\Sigma)$, $L, u$ such that $L \preceq \Sigma \preceq uI$, $t \in \mathbb{N}^+$, $\rho_{1\ldots t}, \beta > 0$
    **Output:** A $(\sum_{i=1}^{t}\rho_i)$-zCDP estimate of $\Sigma$

1:  **procedure** $\text{MVCR}\text{EC}(X_{1\ldots n}, L, u, t, \rho_{1\ldots t}, \beta)$
2:     Let $A_0 = \frac{1}{\sqrt{u}}I, L_0 = \frac{1}{\sqrt{u}}L$.
3:     **for** $i \in [t-1]$ **do**
4:         $(A_i, L_i, Z_i) = \text{MVC}(X_{1\ldots n}, A_{i-1}, L_{i-1}, \rho_i, \frac{\beta}{4(t-1)})$.
5:     **end for**
6:     $(A_t, L_t, Z_t) = \text{MVC}(X_{1\ldots n}, A_{t-1}, L_{t-1}, \rho_t, \frac{\beta}{4})$.
7:     **return** $A_{t-1}^{-1} Z_t A_{t-1}^{-1}$.
8: **end procedure**

---

of an empirical covariance statistic is bounded using the following lemma. [KLSU19] proved a similar statement without an explicit constant, but the optimal constant is important in practice.

**Lemma 3.2.** *Let* $f(D) = \frac{1}{n}\sum_i D_i D_i^T$, *where* $\|D_i\|_2^2 \leq T$. *Then the* $\ell_2$-*sensitivity of* $f$ *(i.e.,* $\max_{D,D'} \|f(D) - f(D')\|_F$, *where* $D$ *and* $D'$ *are neighbors) is at most* $\frac{\sqrt{2}T}{n}$.

Applying the Gaussian mechanism (à la [DTTZ14]) in combination with this sensitivity bound, we again get a private point estimate $Z$ for the covariance, and can also derive inner and outer confidence ellipsoids. This time we require more sophisticated tools, including confidence intervals for the spectral norm of both a symmetric Gaussian matrix and the empirical covariance matrix of Gaussian data. Using valid confidence intervals ensures accuracy, and a sufficiently large $n$ again results in a constant factor squeezing of the ellipsoids, guaranteeing progress.

Putting together the analysis leads to the following theorem.

**Theorem 3.3.** *MVCR*EC *is* $(\sum_{i=1}^{t}\rho_i)$-*zCDP. Furthermore, suppose* $X_1, \ldots, X_n \sim N(0,\Sigma)$, *where* $I \preceq \Sigma \preceq KI$, *and* $n = \tilde{\Omega}((\frac{d^2}{\alpha^2} + \frac{d^2}{\alpha\sqrt{\rho}} + \frac{\sqrt{d^3\log K}}{\sqrt{\rho}}) \cdot \log\frac{1}{\beta})$. *Then MVCR*EC$(X_1, \ldots, X_n, I, K, t =$

$O(\log K), \frac{\rho}{2(t-1)}, \ldots, \frac{\rho}{2(t-1)}, \frac{\rho}{2}, \beta)$ *will return* $\hat{\Sigma}$ *such that* $\|\hat{\Sigma}^{-1/2}\Sigma\hat{\Sigma}^{-1/2} - I\|_F \leq \alpha$ *with probability at least* $1 - \beta$.

# 4   Experimental Evaluation

Before we proceed with our experimental evaluation, we recall the parameters which will be of interest and varied throughout our experiments. We use $n$ for the number of samples, $d$ for the dimension of the data, $R$ for a bound on $\|\mu\|_2$, and $K$ for a bound such that $I \preceq \Sigma \preceq KI$. $t$ is a hyperparameter that represents the number of steps for our iterative algorithms; note that $t = 1$ corresponds to the naïve algorithm mentioned in Section 1.2. Our comparisons will be for the notion of $\rho$-zCDP, for various values of $\rho$. If we are running an algorithm that gives $(\varepsilon, 0)$-DP, we convert the guarantees to $\rho$-zCDP for $\rho = \frac{1}{2}\varepsilon^2$ to make a direct comparison. Code for our algorithms and experiments is provided at https://github.com/twistedcubic/coin-press.

## 4.1   Mean Estimation Experiments

In this section, we present our experimental results on multivariate mean estimation. At a high level, there are two general approaches for the multivariate problem. The first is to solve the univariate problem in each dimension, and combine these results in the natural way. As shown in [KLSU19], with an appropriate setting of parameters, an asymptotically optimal algorithm for the univariate problem leads to an asymptotically near-optimal algorithm for the multivariate problem. The other class of approaches is to work directly in the multivariate space, as done in our novel method MVMREC (Figure 2). We compare the following approaches, labeled with their names as in the legend of our plots:

1. The non-private empirical mean (`Non-private`);

2. Univariate naïve method applied coordinatewise (`Naive coordinatewise`);

3. Karwa-Vadhan [KV18] applied coordinatewise (`KV`);

4. SYMQ of Du et al. [DFM+20] applied coordinatewise (`SYMQ`);

5. Multivariate naïve method (i.e., Algorithm 2 with $t = 1$) (`t = 1`);

6. Algorithm 2 for various $t > 1$ (`t = ♠`, for integer ♠ > 1).

**Implementation Details.**   We use our own implementation of these algorithms except for SYMQ, for which we use the code that accompanies the paper [DFM+20]. There are a number of small tuning details that affect performance in practice – we highlight these changes for our method, and direct the curious reader to our accompanying code for more details. Our algorithm has essentially four hyperparameters: choice of $t$, splitting the privacy budget, radius of the clipping ball, and radius of the confidence ball. We explore the role of $t$ in our experiments. We found that assigning most of the privacy budet to the final iteration increased performance, namely $3\rho/4$ going to the final iteration and $\rho/4(t-1)$ to every other step. In theory, we use a relatively large value for the clipping threshold because it is more convenient for the analysis. In practice, we use a smaller clipping threshold to reduce sensitivity (partially driven by the high-dimensional geometry), which we find improves practical performance.

**Experimental Setup.**  We describe our setup for all the following experiments (later high-lighting any relevant deviations). We generate a dataset of $n$ samples from a $d$-dimensional Gaussian $N(0, I)$, where we are promised the mean is bounded in $\ell_2$-distance by $R$. We run all the methods being compared to ensure a guarantee of $\rho$-CDP. We run each method 100 times, and report the trimmed mean, with trimming parameter 0.1. We trim because a single failure can significantly inflate the average error. Our plots display the $\ell_2$-error of a method on the y-axis. In some cases, we focus on the excess $\ell_2$-error over the non-private baseline to provide a clearer comparison of private methods. We did not focus on optimizing the running time, but most of the plots (each involving about 1,000 runs of our algorithm) took only a few minutes to generate on a laptop computer with an Intel Core i7-7700HQ CPU.

**A Note on Hyperparameters.**  We note that in different experiments, the best variant of our mechanism corresponded to different numbers of iterations, ranging from $t = 2$ to $t = 10$ iterations. However, we observe that in all experiments $t = 10$ performs competitively with the best choice of $t$ for that setting, showing that the method is relatively robust to how this hyperparameter is tuned. Moreover, since the final error is ultimately determined by the value of $R$ used in the final call to the one-step estimator, and the sequence of value $R$ follows a deterministic recurrence, one could add a data-independent preprocessing step to evaluate the recurrence for various values of $t$ and determine the optimal choice. No other hyperparameters tuning was used between different experiments.

### 4.1.1   Results and Discussion

Figure 3: Baseline comparison of mean estimation algorithms ($d = 50, R = 10\sqrt{d}, \rho = 0.5$). Our algorithm with $t = 2$ outperforms all other methods.

In our first experiment (Figure 3), we consider estimation in $d = 50$ dimensions with $\rho = 0.5$. We set the initial radius to $R = 10\sqrt{d}$, which roughly means that the user can estimate the mean of each coordinate a priori to within $\pm 10$ standard deviations. We then measure the error with varying choices of sample size $n$ between $10^3$ and $10^4$. The first and second panels show that our method (with $t = 2$ iterations) significantly outperforms previous methods, and offers error that is quite close to the non-private error. Concretely, we see that the additional cost of privacy is about 27% for $n = 1,000$ and decreases to just 2% for $n = 10,000$.

In our next experiment (Figure 4), we consider the effect of increasing the initial radius $R$, which corresponds to a user with less a priori knowledge of the parameters. Here, we fix $d = 50, n = 1,000, \rho = 0.5$ and vary $R$. We can see that when our method is run with $t = 10$ iterations, its error is essentially independent of $R$, showing no visible change in error, even when we increase $R$ by three orders of magnitude. In contrast, all other methods show dramatically

Figure 4: Effect of increasing $R$ ($d = 50, n = 1000, \rho = 0.5$). Even when SYMQ is advantaged by giving it $n = 2000$ samples, our method with $t = 10$ outperforms it for large $R$.

worse performance as $R$ grows. We note that, as predicted by theoretical analysis, the error of SYMQ appears to observe a threshold behavior, in which the error grows proportionally to $R$ when $n$ goes below this threshold and is nearly independent of $R$ when $n$ is above the threshold. Thus, in the final panel, we "advantage" SYMQ by giving it twice as many samples ($n = 2000$ instead of $n = 1000$), and we observe that our method continues to have somewhat lower error.

Figure 5: Low dimensional comparison ($d = 2, R = 10\sqrt{d}, \rho = 0.5$). Our method is superior even in only 2 dimensions.

Figure 6: High dimensional comparison ($d = 500, R = 10\sqrt{d}, \rho = 0.5$). Our method is effective for all values of $n$.

In the next set of experiments (Figures 5 and 6) we consider both larger dimension ($d = 500$) and smaller dimension ($d = 2$). In both cases we consider $R = 10\sqrt{d}$ and $\rho = 0.5$ and very $n$. For bivariate data (Figure 5), our method (with $t = 2$ iterations) still has the best performance, while other methods have error comparable to the naïve algorithm. For large dimension (Figure 6), SYMQ is ineffetive at small sample sizes while our method (with $t = 2$) competes well with non-private estimation—even with $n < 4d$ samples, the cost of privacy is less than a factor of 2.

Next (Figure 7), we consider the effect of varying the privacy parameter $\rho$. We fix choices of the other parameters ($d = 50, n = 2000, R = 10\sqrt{d}$). We observe that our methods remain

Figure 7: Varying the privacy level ($d = 50, n = 2000, R = 10\sqrt{d}$). Our methods are effective even at high privacy levels (small $\rho$).

superior for all choices of $\rho$, while coordinatewise methods are ineffective for high levels of privacy. In particular, for these parameters, our the cost of privacy for our method remains smaller than a factor of 2, even with privacy levels as low as $\rho = 0.04$, which roughly corresponds to $\varepsilon = 0.25$.

Lastly (Figure 8), we give a proof-of-concept showing that our methods do not strictly require Gaussian data. We fix $d = 50, R = 10\sqrt{d}, \rho = 0.5$ and $t = 2$ iterations, and consider data drawn from various distributions with heavier tails: the multivariate Laplace and the multivariate Student's $t$-distribution with 3 degrees of freedom. We plot the excess error compared to non-private estimation.[6] Even with model misspecfciation, our methods remain effective.

Figure 8: Mean estimation with non-Gaussian data ($d = 50, R = 10\sqrt{d}, \rho = 0.5, t = 2$). Our method is still effective even under data with heavier tails.

### 4.2 Covariance Estimation Experiments

We present our experimental results on multivariate covariance estimation. Covariance estimation offers fewer approaches to compare with, as it is unclear how to apply a univariate algorithm to the multivariate setting. In particular, estimating the off-diagonal terms of the covariance matrix is a very different problem than estimating the variance of a single normal, so an entrywise approach will run into significant challenges unless we assume the covariance matrix is diagonal. Thus, we compare the following two approaches:

1. Naïve method (MVCREC with $t = 1$)[7];

2. Our method (MVCREC for various $t > 1$).

We do not compare with the algorithm of [KLSU19]. When implementing their algorithm, we found it too difficult to tune the numerous intertwined hyperparameters well enough to produce non-trivally accurate estimates. However, we note that our method can be seen as a "smoother" variant of their approach that is easier to implement and tune.

**Implementation Details.**   As before, there are four hyperparameters: choice of $t$, splitting of the privacy budget, radius of the clipping ball, and radius of the confidence ball. We find that the same optimizations to the choice of $t$ and the division of the privacy budget $\rho$ are helpful for covariance estimation. However, for covariance estimation we also find that an even more aggressive shrinking of the confidence ellipsoids gives the best concrete performance.

### 4.2.1   Synthetic Data Experiments

**Experimental Setup.**   We describe our setup for the following experiments, highlighting any relevant deviations later. We generate a dataset of $n$ samples from a $d$-dimensional Gaussian, either $N(0, I)$ (the isotropic case), or $N(0, \Sigma)$, where $\Sigma$ is a random rotation of a matrix with $d/2$ eigenvalues equal to each of the values $K$ and $1$ (the skewed case). We run the methods being compared to ensure a guarantee of $\rho$-CDP. As with mean estimation, we run 100 trials of the algorithm and report the trimmed mean with trimming level 0.1. Our plots display the Mahalanobis error of a method on the y-axis, or, equivalently, the Frobenius error after accounting for differences in scaling in all directions. All experiments were completed within a few minutes on a laptop computer with an Intel Core i7-7700HQ CPU.

**Results and Discussion.**   In our first set of experiments (Figures 9 and 10) we consider covariance estimation in $d = 10$ dimensions with $\rho = 0.5$. Note that a covariance matrix with $d = 10$ dimensions has 55 non-redundant parameters, so this setting is somewhat analogous to our experiments with mean estimation in $d = 50$ dimensions.

Figure 9: Baseline comparison of covariance estimation algorithms ($d = 10, K = 10\sqrt{d}, \rho = 0.5$) with isotropic covariance. Our algorithm significantly outpforms the naïve baseline for all $t > 1$.

For the isotropic case (Figure 9) we can see that our method significantly outperforms the naïve baseline for all choices of $t > 1$, with $t = 3$ iterations giving the best results for these parameters. For this setting of $t = 3$, the cost of privacy is within a factor of 1.5 for $n = 3000$.

Figure 10: Baseline comparison of covariance estimation algorithms ($d = 10, K = 10\sqrt{d}, \rho = 0.5$) with skewed covariance. For $t = 2$ our algorithm is competitive with the non-private ideal.

For the skewed case (Figure 10) we see that with $t = 2$ our algorithm is significantly better than the non-private baseline, although now accuracy degrades for larger choices of $t$.

Figure 11: Effect of increasing $K$ ($d = 10, n = 7000, \rho = 0.5$) with isotropic covariance (top) and skewed covariance (bottom).

In our next set of experiments (Figure 11) we investigate the effect of the initial radius $K$, fixing other parameters ($d = 10, n = 7000, \rho = 0.5$). We can see that for both the isotropic (top) and skewed case (bottom) the $t = 1$ baseline is the least accurate of all alternatives. Our methods perform significantly better for values of $t = 3, 4, 5$, although the optimal choice varies.

Figure 12: Low dimensional comparison ($d = 2, K = 10\sqrt{d}, \rho = 0.5$) with isotropic covariance (top) and skewed covariance (bottom).

Next we experiment with the low-dimensional case of bivariate data (Figure 12). We set $d = 2$ and vary $n$ while holding other parameters fixed ($K = 10\sqrt{d}, \rho = 0.5$). For the isotropic

case (top) we see that all methods significantly outperform the naïve baseline, and for the skewed case (bottom) our method significantly outperforms the naïve baseline for $t = 2$ iterations, but is significantly more sensitive to the choice of $t$. As one would expect, since the dimension is small, the cost of privacy is quite low for dataset sizes around $n = 1000$.

Figure 13: High dimensional comparison ($d = 100, K = 10\sqrt{d}, \rho = 0.5$) with isotropic covariance. $t > 1$ substantially outperform the $t = 1$ baseline, and $t = 3$ is competitive with the non-private baseline up to a small constant factor for larger $n$.

Similarly, we perform a comparison for the high-dimensional case (Figure 13), setting $d = 100$, corresponding to 5050 non-redundant parameters. We vary $n$ and fix the other parameters ($K = 10\sqrt{d}, \rho = 0.5$).

Figure 14: Varying the privacy level ($d = 10, n = 8000, K = 10\sqrt{d}$) with isotropic covariance (top) and skewed covariance (bottom).

Finally, we investigate the role of the privacy budget $\rho$ (Figure 14), fixing other parameters $(d = 10, K = 10\sqrt{d}, n = 8000)$. In the isotropic case (top) we see that larger $t$ perform better, particularly as $\rho$ becomes very small. In the skewed case (bottom), we see that $t = 2$ is the most accurate, while larger $t$ seem to perform poorly at small privacy budgets.

**A Note on Hyperparameters.** As with our mean estimation, our covariance estimator has a hyperparameter $t$ specifying the number of iterations. Compared to mean estimation, the error of the estimator appears to be much more sensitive to tuning this hyperparameter. Moreover, we do not know how to predict the best choice of hyperparameter in a data-independent way. Understanding how to set this hyperparameter privately, or modify the algorithm to make it less sensitive to it, is an important direction for future study.

## 4.3   Map of Europe

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

[3]Formally, $(\varepsilon, 0)$-DP $\implies \frac{1}{2}\varepsilon^2$-zCDP $\implies (\varepsilon\sqrt{2\log(1/\delta)} + \frac{1}{2}\varepsilon^2, \delta)$-DP for every $\delta > 0$

[4]Given two probability distributions $P, Q$ over $\Omega$, $D_\alpha(P\|Q) = \frac{1}{\alpha-1}\log\left(\sum_x P(x)^\alpha Q(x)^{1-\alpha}\right)$.

[5]The astute reader will notice that the algorithm itself never uses the lower bound on the covariance matrix. However, it is useful to keep in mind for intuition, and will come up in the analysis.

[6]Note that, since we compare with non-private estimation on independent datasets, the excess error can sometimes be negative due to randomness in the simulation, though it is always non-negative on average.

[7]This method, which amounts to clipping the data and then adding noise to the empirical covariance, is sometimes called "Analyze Gauss," due to its use in [DTTZ14].

[8]https://github.com/NovembreLab/Novembre_etal_2008_misc

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

# A  New Algorithms for Univariate Gaussian Parameter Estimation

In this section, we present our algorithms for estimating the mean and variance of a univariate Gaussian. We will write all our algorithms to give zCDP privacy guarantees, but the same approach can give pure DP algorithms in the univariate setting. One must simply switch Gaussian to Laplace noise, swap in the appropriate tail bounds for the confidence interval, and apply basic composition rather than zCDP composition.

## A.1  Univariate Private Mean Estimation

We start with our univariate private mean estimation algorithm UVMRec (Algorithm 6). The guarantees are presented in Theorem A.1, note that the sample complexity is optimal in all parameters up to logarithmic factors [KV18]. While our algorithm and results are stated for a Gaussian with known variance, the same guarantees hold if the algorithm is only given the true variance up to a constant factor. Algorithm 6 is an iterative invocation of Algorithm 5, each step of which makes progress by shrinking our confidence interval for where the true mean lies. This is the simplest instantiation of our general algorithmic formula. Additionally, our proof of correctness is spelled out in full detail for this case – as the other proofs follow an almost identical structure, we only describe the differences.

**Theorem A.1.** UVMRec *is* $(\sum_{i=1}^{t} \rho_i)$*-zCDP. Furthermore, suppose we are given samples* $X_1, \ldots, X_n$ *from* $N(\mu, \sigma^2)$*, where* $|\mu| < R\sigma$ *and* $n = \tilde{\Omega}\Big(\Big(\frac{1}{\alpha^2} + \frac{1}{\alpha\sqrt{\rho}} + \frac{\sqrt{\log R}}{\sqrt{\rho}}\Big) \cdot \log(1/\beta)\Big)$*. Then*

---

**Algorithm 5** One Step Private Improvement of Mean Interval

---

      **Input:** $n$ samples $X_{1...n}$ from $N(\mu, \sigma^2)$, $[\ell, r]$ containing $\mu$, $\sigma^2$, $\rho_s, \beta_s > 0$

      **Output:** A $\rho_s$-zCDP interval $[\ell', r']$

1: **procedure** UVM$(X_{1...n}, \ell, r, \sigma^2, \rho_s, \beta_s)$

2:       Project each $X_i$ into the interval $[\ell - \sigma\sqrt{2\log(2n/\beta_s)}, r + \sigma\sqrt{2\log(2n/\beta_s)}]$.

3:       Let $\Delta = \frac{r - \ell + 2\sigma\sqrt{2\log(2n/\beta_s)}}{n}$.

4:       Compute $Z = \frac{1}{n}\sum_i X_i + Y$, where $Y \sim N\left(0, \left(\frac{\Delta}{\sqrt{2\rho_s}}\right)^2\right)$.

5:       **return** the interval $Z \pm \sqrt{2\left(\frac{\sigma^2}{n} + \left(\frac{\Delta}{\sqrt{2\rho_s}}\right)^2\right)\log(2/\beta_s)}$.

6: **end procedure**

---

---

**Algorithm 6** Private Confidence-interval-based Univariate Mean Estimation

---

      **Input:** $n$ samples $X_{1...n}$ from $N(\mu, \sigma^2)$, $[\ell, r]$ containing $\mu$, $\sigma^2$, $t \in \mathbb{N}^+$, $\rho_{1...t}, \beta > 0$

      **Output:** A $(\sum_{i=1}^{t} \rho_i)$-zCDP estimate of $\mu$

1: **procedure** UVMREC$(X_{1...n}, \ell, r, \sigma^2, t, \rho_{1...t}, \beta)$

2:     Let $\ell_0 = \ell, r_0 = r$.

3:     **for** $i \in [t-1]$ **do**

4:         $[\ell_i, r_i] = \text{UVM}(X_{1...n}, \ell_{i-1}, r_{i-1}, \sigma^2, \rho_i, \beta/4(t-1))$.

5:     **end for**

6:     $[\ell_t, r_t] = \text{UVM}(X_{1...n}, \ell_{t-1}, r_{t-1}, \sigma^2, \rho_t, \beta/4)$.

7:     **return** the midpoint of $[\ell_i, r_i]$.

8: **end procedure**

---

UVMREC$(X_1, \ldots, X_n, -R, R, \sigma^2, t = O(\log R), \rho/2(t-1), \ldots, \rho/2(t-1), \rho/2, \beta)$ *will return* $\hat{\mu}$ *such that* $|\mu - \hat{\mu}| \leq \alpha\sigma$ *with probability at least* $1 - \beta$.

*Proof.* We start by proving privacy. Observe that by application of the Gaussian mechanism (Lemma 2.6) in Line 4 of Algorithm 5, this algorithm is $\rho$-zCDP. Privacy of Algorithm 6 follows by composition of zCDP (Lemma 2.4).

We start by analyzing the $t - 1$ iterations of the Line 3, each of which calls Algorithm 5. We prove two properties of this algorithm. Informally, it will always create a valid confidence interval, and the confidence interval shrinks by a constant factor. More formally:

1. First, if Algorithm 5 is invoked with $\mu \in [\ell, r]$, then it returns an interval $[\ell', r'] \ni \mu$, with probability at least $1 - 2\beta_s$. To show this, we begin by considering a variant of the algorithm where Line 2 is omitted. In this case, observe that $Z$ is a Gaussian with mean $\mu$ and variance $\frac{\sigma^2}{n} + \left(\frac{\Delta}{\sqrt{2\rho_s}}\right)^2$. Then $\mu \in [\ell', r']$ with probability $1 - \beta_s$ by Fact C.1. Re-introducing Line 2, Fact C.1 and a union bound imply that the total variation distance between the true process and the one without Line 2 is at most $\beta_s$, and thus $\mu \in [\ell', r']$ with probability at least $1 - 2\beta_s$.

2. Second, if $r - \ell > C\sigma$ for some absolute constant $C$, then $r' - \ell' \leq \frac{1}{2}(r - \ell)$. The width of the interval $r' - \ell' = 2\sqrt{2\left(\frac{\sigma^2}{n} + \left(\frac{\Delta}{\sqrt{2\rho_s}}\right)^2\right)\log(2/\beta_s)} \leq 2\sqrt{2\log(2/\beta_s)}\left(\frac{\sigma}{\sqrt{n}} + \frac{\Delta}{\sqrt{2\rho_s}}\right)$. The former term can be bounded as $O(1) \cdot \sigma$, using the facts that $\beta_s = \beta/4(t-1)$, $t = O(\log R)$, and $n = \Omega(\log(\log R/\beta))$. We rewrite and bound the latter term as $O\left(\frac{\sqrt{\log R}\sqrt{\log(n \log R/\beta)}(r - \ell + \sigma)}{n\sqrt{\rho}}\right)$, and the claim follows based on our condition on $n$.

We turn to the final call, in Line 6. The idea will be that the interval $[\ell_{t-1}, r_{t-1}]$ will now be so narrow (after the previous shrinking) that the noise addition in Algorithm 5 will be insignificant. Invoking the two points above respectively, we have that: (1) $\mu \in [\ell_{t-1}, r_{t-1}]$ with probability at least $1 - \beta/2$ (where we used a union bound), and (2) $|r_{t-1} - \ell_{t-1}| \leq O(1) \cdot \sigma$ (where we also used $t = O(\log R)$). Conditioning on these, we show that $|Z - \mu| \leq \alpha\sigma$. Similar to before, we consider a variant of Algorithm 5 where Line 2 is omitted, and by Gaussian tail bounds,

$$|Z - \mu| \leq O\left(\sqrt{\left(\frac{\sigma^2}{n} + \left(\frac{\sigma\sqrt{\log(n/\beta)}}{n\sqrt{\rho}}\right)^2\right)\log(1/\beta)}\right) \text{ with probability at least } 1 - \beta/4. \text{ Our choice}$$

of $n$ bounds this expression by $\alpha\sigma$. Observing that (similar to before) Line 2 only rounds any point with probability at most $\beta/4$, the estimate is accurate with probability at least $1 - \beta/2$. Combining with the previous $\beta/2$ probability of failure completes the proof. $\square$

## A.2 Univariate Private Variance Estimation

We proceed to present our our univariate private variance estimation algorithm UVVREC (Algorithm 8). The guarantees are presented in Theorem A.2. Note that our algorithms work given an arbitrary interval $[\ell, u]$ containing $\sigma^2$, but for simplicity, we normalize so that $\ell = 1$ and then $u = K$. The first two terms in the sample complexity are optimal up to logarithmic factors, though using different methods, the third term's dependence on $K$ can be reduced from $\sqrt{\log K}$ to $\sqrt{\log \log K}$ [KV18]. As our primary focus in this paper is on providing simple algorithms requiring minimal hyperparameter tuning, we do not attempt to explore optimizations for this term. Our

algorithm will take as input samples $X_i \sim N(0, \sigma^2)$ from a zero-mean Gaussian. One can easily reduce to this case from the general case: given $Y_i \sim N(\mu, \sigma^2)$, then $\frac{1}{\sqrt{2}}(Y_{2i-1} - Y_{2i}) \sim N(0, \sigma^2)$.

---

**Algorithm 7** One Step Private Improvement of Variance Interval

**Input:** $n$ samples $X_1, \ldots, X_n$ from $N(0, \sigma^2)$, $[\ell, u]$ containing $\sigma^2$, $\rho_s, \beta_s > 0$
**Output:** A $\rho_s$-zCDP interval $[\ell', u']$

1: **procedure** UVV$(X_{1\ldots n}, \ell, u, \rho_s, \beta_s)$
2:     Compute $W_i = X_i^2$.                                   $\triangleright\ W_i \sim \sigma^2 \chi_1^2$
3:     Project each $W_i$ into $[0, u \cdot (1 + 2\sqrt{\log(1/\beta_s)} + 2\log(1/\beta_s))]$.
4:     Let $\Delta = \frac{1}{n} \cdot u \cdot (1 + 2\sqrt{\log(1/\beta_s)} + 2\log(1/\beta_s))$.
5:     Compute $Z = \frac{1}{n}\sum_i W_i + Y$, where $Y \sim N\left(0, \left(\frac{\Delta}{\sqrt{2\rho_s}}\right)^2\right)$.
6:     **return** the intersection of $[\ell, u]$ with the interval

$$Z + 2u \cdot \left[ -\frac{\Delta}{\sqrt{\rho_s}}\sqrt{\log(4/\beta_2)} - \sqrt{\frac{\log(4/\beta_2)}{n}} - \frac{\log(4/\beta_2)}{n}, \frac{\Delta}{\sqrt{\rho_s}}\sqrt{\log(4/\beta_2)} + \sqrt{\frac{\log(4/\beta_2)}{n}} \right].$$

7: **end procedure**

---

**Algorithm 8** Private Confidence-Interval-Based Univariate Variance Estimation

**Input:** $n$ samples $X_{1\ldots n}$ from $N(0, \sigma^2)$, $[\ell, u]$ containing $\sigma^2$, $t \in \mathbb{N}^+$, $\rho_{1\ldots t}, \beta > 0$
**Output:** A $(\sum_{i=1}^t \rho_i)$-zCDP estimate of $\sigma^2$

1: **procedure** UVVREC$(X_{1\ldots n}, \ell, u, t, \rho_{1\ldots t}, \beta)$
2:     Let $\ell_0 = \ell, u_0 = u$.
3:     **for** $i \in [t-1]$ **do**
4:         $[\ell_i, u_i] = $ UVV$(X_{1\ldots n}, \ell_{i-1}, u_{i-1}, \rho_i, \beta/4(t-1))$.
5:     **end for**
6:     $[\ell_t, u_t] = $ UVV$(X_{1\ldots n}, \ell_{t-1}, u_{t-1}, \rho_t, \beta/4)$.
7:     **return** the midpoint of $[\ell_i, u_i]$.
8: **end procedure**

---

**Theorem A.2.** UVVREC *is $(\sum_{i=1}^t \rho_i)$-zCDP. Furthermore, suppose we are given i.i.d. samples $X_1, \ldots, X_n$ from $N(0, \sigma^2)$, where $1 \le \sigma^2 < K$ and $n = \tilde{\Omega}\left(\left(\frac{1}{\alpha^2} + \frac{1}{\alpha\sqrt{\rho}} + \frac{\sqrt{\log K}}{\sqrt{\rho}}\right) \cdot \log(1/\beta)\right)$. Then UVVREC$(X_1, \ldots, X_n, 1, K, t = O(\log K), \rho/2(t-1), \ldots, \rho/2(t-1), \rho/2, \beta)$ will return $\hat{\sigma}^2$ such that $|\hat{\sigma}^2/\sigma^2 - 1| \le \alpha$ with probability at least $1 - \beta$.*

*Proof.* Overall, the proof is very similar to that of Theorem A.1, so we only highlight the differences. First, we note that the proof of privacy is identical, via the Gaussian mechanism and composition of zCDP.

    We again analyze each of the calls in the loop, which this time call Algorithm 7. First, if Algorithm 7 is invoked with $\sigma^2 \in [\ell, u]$, then it returns an interval containing $\sigma^2$ with probability at least $1 - 2\beta_s$. We use the same argument as before: in Line 3, we observe that the $W_i$'s are scaled chi-squared random variables with 1 degree of freedom and apply Fact C.2. In Line 6, our confidence interval is generated using a combination of Facts C.1 and C.2. Note that, since

the true variance is unknown, we conservatively scale by the upper bound on the variance $u$, to guarantee that the confidence interval is valid.

Second, we argue that we make "progress" each step. Specifically, we claim that if $u/\ell \geq C$ for some absolute constant $C \gg 1$, then we return an interval of width $\leq \frac{1}{2}(u - \ell)$. The condition implies that the width of the interval starts at $u - \ell \geq u(1 - 1/C)$. Substituting our condition on $n$ into the width of the confidence interval, we can upper bound its width by $C'u$, where $0 < C' \ll 1$ is a constant that can be taken arbitrarily close to 0 based on the hidden constant in the condition on $n$. Combining these two facts yields the claim.

Now, similar to before, we inspect the final call to UVV in Line 6 of UVVREC. Again, due to the second claim above and the fact that we chose $t = \log_2 K$, this will be called with $\ell$ and $u$ such that $u_{t-1}/\ell_{t-1} \leq C$. The first claim above implies that $\ell_{t-1} \leq \sigma^2 \leq r_{t-1}$ with probability at least $1 - \beta/2$ (which we condition on). As argued above, the confidence interval defined in Line 6 contains $\sigma^2$ with probability at least $1 - \beta/2$. Using the theorem's condition on $n$ (with a sufficiently large hidden constant), we have that its width is at most $\alpha u/C \leq \alpha\ell \leq \alpha\sigma^2$, which implies the desired conclusion. $\qquad\square$

# B  Missing Proofs from Section 3

## B.1  Proof of Theorem 3.1

The proof is very similar to that of Theorem A.1, so we assume familiarity with that and only highlight the differences. First, we note that the proof of privacy is identical, via the Gaussian mechanism and composition of zCDP.

We again analyze each of the calls in the loop, which this time refer to Algorithm 1. First, if Algorithm 1 is invoked with $\mu \in B_2(c, r)$, then it returns a ball $B_2(c', r') \ni \mu$ with probability at least $1 - \beta_1 - \beta_2$. The argument is identical to before, but this time using a tail bound for a multivariate Gaussian (Fact C.2) instead of the univariate version. Second, if $r > C\sqrt{d}$ for some constant $C > 0$, then $r' < r/2$. Once again, this can reasoned by inspecting the expression for $r'$ and applying our condition on $n$ (in particular, focusing on the term which is $\tilde{\Omega}\left(\frac{\sqrt{d \log R}}{\sqrt{\rho}}\right)$).

Finally, we inspect the last call in Line 6. Similar to before, the radius of the ball $r_{t-1} \leq C\sqrt{d}$. In this final call, we can again couple the process with the one which doesn't round the points to the ball (which we will focus on), where the probability that the two processes differ is at most $\beta/4$. By Fact C.2, we have that $\|Z - \mu\|_2 \leq \tilde{O}\left(\sqrt{\left(\frac{1}{n} + \frac{d}{n^2\rho}\right)}\sqrt{d + \sqrt{d \log(1/\beta_2)} + \log(1/\beta_2)}\right)$ with probability at least $1 - \beta/4$. Substituting in our condition on $n$ and accounting for the failure probability at any previous step completes the proof.

## B.2  Proof of Lemma 3.2

Suppose the datasets differ in that one contains a point $X$ which is replaced by the point $Y$ in the other dataset.

$$
\begin{aligned}
\left\| \frac{1}{n}\left(XX^T - YY^T\right) \right\|_F &= \frac{1}{n}\sqrt{\mathrm{Tr}((XX^T - YY^T)^2)} \\
&= \frac{1}{n}\sqrt{\mathrm{Tr}((XX^T)^2 - XX^TYY^T - YY^TXX^T + (YY^T)^2)} \\
&\leq \frac{1}{n}\sqrt{\|XX^T\|_F^2 + \|YY^T\|_F^2} \\
&= \frac{1}{n}\sqrt{\|X\|_2^4 + \|Y\|_2^4} \\
&\leq \frac{1}{n}\sqrt{2T^2} \\
&= \frac{\sqrt{2}}{n}T
\end{aligned}
$$

The first inequality is since $\mathrm{Tr}(AB) \geq 0$, for any positive semi-definite matrices $A$ and $B$.

## B.3  Proof of Theorem 3.3

Privacy again follows from the Gaussian mechanism and composition of zCDP. Note that this time, the sensitivity bound is not obvious – the analysis depends on Lemma 3.2.

Next, we analyze each of the calls in the loop, referring to Algorithm 3. If Algorithm 3 is invoked with $L \preceq A\Sigma A \preceq I$, then it returns $A', L'$ such that $L' \preceq A'\Sigma A' \preceq I$ with probability at least $1 - \beta_1 - \beta_2$. The analysis is similar to before: we bound the probability that a point gets adjusted in Line 3 using Fact C.2. We bound the spectral norm of the error due to sampling and the Gaussian noise matrix using Lemma C.3 and C.4, respectively. The claim follows from elementary linear algebra: the above spectral norm bounds imply that $Z - (\eta + \nu)I \preceq A\Sigma A \preceq Z + \eta I$, and scaling by $(Z + \eta I)^{-1/2}$ gives the result.

We now argue that we make progress each step. For all unit vectors $v$, if $v^T L v \leq C$ for some absolute constant $0 < C < 1/2$, then $v^T \tilde{U} v - v^T(Z - (\eta + \nu)I)v \leq \frac{1}{2}(1 - v^T L v)$. To motivate the conclusion of this claim: we note that the two matrices $\tilde{U}$ and $(Z - (\eta + \nu)I)$ are the matrices $I$ and $L'$ scaled by $\tilde{U}^{1/2}$ – thus, we will argue that, in any direction, the width of the window bounding $v^T A\Sigma A v$ shrinks by a factor of at least 2. Returning to the proof: the condition $v^T L v \leq C$ implies that that $1 - v^T L v \geq 1 - C$. We note that $v^T(Z + \eta I)v - v^T(Z - (\eta + \nu)I)v = 2\eta + \nu$. Substituting our condition on $n$ will upper bound $2\eta + \nu \leq C'$, for some absolute constant $C' \ll \frac{1}{2}(1 - C) \leq \frac{1}{2}(1 - v^T L v)$, as desired.

Now, similar to before, we inspect the final call in Line 6. The second claim above, unrolling the effect of the scaling matrices $A$, and the fact that we chose $t = O(\log K)$ implies that $v^T L_{t-1} v \geq C$. The first claim above implies that $L_{t-1} \preceq A_{t-1}\Sigma A_{t-1} \preceq I$ with probability at least $1 - \beta/2$ (which we condition on). With these in place, analysis follows similarly to Lemma 3.6 of [KLSU19]. Sketching the argument: our condition on $n$ implies that the Frobenius norm of both the empirical covariance and the noise added will be bounded by $\alpha$. Rescaling $Z$ by the scaling matrix $A_{t-1}$ gives the desired result.

# C   Concentration Inequalities and Tail Bounds

The following tail bounds are standard.

**Fact C.1.** *If $X \sim N(\mu, \sigma^2)$, then $\Pr(|X - \mu| \geq \sigma\sqrt{2\log(2/\beta)}) \leq \beta$.*

**Fact C.2** (Lemma 1 of [LM00]). *If $X$ is a chi-squared random variable with $k$ degrees of freedom, then $\Pr(X - k \geq 2\sqrt{k\log(1/\beta)} + 2\log(1/\beta)) \leq \beta$ and $\Pr(k - X \geq 2\sqrt{k\log(1/\beta)}) \leq \beta$. Thus, if $Y \sim N(0, I)$, then $\Pr(\|Y\|_2^2 \geq d + 2\sqrt{d\log(1/\beta)} + 2\log(1/\beta)) \leq \beta$.*

We also need the following bound on the spectral error of an empirical covariance matrix.

**Lemma C.3** ((6.12) of [Wai19]). *Suppose we are given $X_1, \ldots, X_n \sim N(0, \Sigma) \in \mathbb{R}^d$, and let $\hat{\Sigma} = \frac{1}{n}\sum_{i=1}^n X_i X_i^T$. Then*

$$\|\hat{\Sigma} - \Sigma\|_2 \leq \|\Sigma\|_2 \left( 2\left( \sqrt{\frac{d}{n}} + \sqrt{\frac{2\ln(\beta/2)}{n}} \right) + \left( \sqrt{\frac{d}{n}} + \sqrt{\frac{2\ln(\beta/2)}{n}} \right)^2 \right)$$

*with probability at least $1 - \beta$.*

Finally, we need a bounds on the spectral norm of a symmetric matrix with random Gaussian entries.

**Lemma C.4.** *Let $Y$ be the $d \times d$ matrix where $Y_{ij} \, N(0, \sigma^2)$ for $i \leq j$, and $Y_{ij} = Y_{ji}$ for $i > j$. Then with probability at least $1 - \beta$, we have the following bound:*

$$\|Y\|_2 \leq \sigma \left( 2\sqrt{d} + 2d^{1/6}\log^{1/3} d + \frac{6(1 + (\log d/d)^{1/3})\sqrt{\log d}}{\sqrt{\log(1 + (\log d/d)^{1/3})}} + 2\sqrt{2\log(1/\beta)} \right)$$

*Proof.* First, we have the following bound on the expectation of the spectral norm:

$$\mathbf{E}\|Y\|_2 \leq \sigma \left( 2\sqrt{d} + 2d^{1/6}\log^{1/3} d + \frac{6(1 + (\log d/d)^{1/3})\sqrt{\log d}}{\sqrt{\log(1 + (\log d/d)^{1/3})}} \right).$$

This is from Theorem 1.1 of [BVH16], fixing the value of $\varepsilon = \frac{\log d}{d}$. The desired tail bound follows since the spectral norm is 2-Lipschitz for this class of symmetric matrices, and by Gaussian concentration of Lipschitz functions (e.g., Proposition 5.34 of [Ver12]). $\qquad\square$