[Reviews · NeurIPS 2020]

Review 1

Summary and Contributions: The submission considers the problem of estimating the mean and covariance matrix of an underlying multivariate sub-Gaussian distribution, given samples from that distribution. While the empirical mean and convariance are optimal estimators, they do not guarantee privacy. The aim of this paper is to devise estimators that offer (concentrated) differential privacy guarantees. The authors present algorithms for the above problem. The mean estimation works iteratively by maintaining a ball which contains the true mean of the distribution with high probability. At each step, the algorithm shrinks the ball by a suitable factor by doing a combination of projection and clipping so that the radii converge fast enough to give an estimate of the mean with desired accuracy. Covariance estimation works similarly, but an assumption is needed regarding bounds on the covariance matrix, i.e., it is assumed that the covariance matrix M satisfies I <= M <= K*I for some K (where <= indicates spectral ordering, i.e., A <= B if B-A is PSD). The authors justify this assumption by noting that users often have a (weak) prior on the mean/covariance of the data. (In theory, it seems that a better prior should be able to improve the number of iterations needed to converge.) Moreover, the paper provides experiments to demonstrate effectiveness of the proposed algorithms. The first experiment is on synthetic data, where data is drawn from a Gaussian distribution. Plots show that for sufficiently large datasets, the error produced by the authors' approach gets close to the error of the non-private empirical mean/covariance estimators. The second experiment tests out the method on PCA, using a dataset of genetic data in Europe.

Strengths: The paper is generally well-written, does a good job of motivating the question and discussing prior/related works, and has a good mix of theory and experiments. The algorithms proposed in the paper make great progress on a natural private analogue of a fundamental question in statistics. The experiments furthermore show the promise of the approach.

Weaknesses: It is not entirely clear to me what the theoretical differences are between this work and [24] (Kamath, Li, Singhal, Ullman). The bounds proved in this work seem virtually identical to the ones in [24]. Their is some allusion to the fact that this work uses a "softer" iterative method, but the comparison is not clear. The paper could be strengthened by having a better discussion of this to make it clear what the conceptual novelty (if any) is in this work compared to [24]. The other weakness concerns some of the choices in the experiment. For instance, for the PCA experiment, there does not seem to be a comparison to the result of using other mean/covariance estimation methods (the only comparison is with the same method using varying values of t). Also, some general intutition about how to set t (and what the tradeoffs are) would have been useful: Some of the baseline comparisons only look at t=1 and t=2, but then other experiments, e.g., effect of increasing R experiment, use values like t=4,10 out of the blue. Moreover, the qualitative difference between t=4 and t=10 in the plot analyzing the effect of increasing R merits some discussion.

Correctness: The claims and method seem correct as far as I can tell, but I have not verified the details which appear in the supplemental information. Experimental methodology seem correct.

Clarity: Yes, I found the paper easy to follow.

Relation to Prior Work: Overall, the discussion of prior work is decent. However, the work should be a bit clearer about where the conceptual/theoretical differences lie in comparison to [24] (see above discussion under Weaknesses).

Reproducibility: Yes

Additional Feedback: N/A


Review 2

Summary and Contributions: This paper proposes two new algorithms for differentially private mean and covariance estimation in the multivariate setting. Previous works have mostly focused on the univariate case. The idea of the proposed algorithms is to work directly in the multivariate and iteratively constraint the search space using smaller and smaller balls for mean estimation, and using two ellipsoids for variance estimation. The paper compares experimentally the proposed algorithms with adaptations to the multivariate case of previous algorithms proposed for the univariate case.

Strengths: -as far as I know, this is the first paper to directly address differentially private mean and variance estimation in the multivariate setting, -the algorithms are relatively simple, but the utility analysis are non-trivial, especially for variance estimation -the experimental evaluation shows clear improvement with respect to using univariate mean and variance estimation methods

Weaknesses: -works in the univariate case have also explored lower bounds, while the paper here focus only on upper bounds -the experimental part is in general good, but it doesn't clarify some important aspect of the algorithm design. Specifically, it doesn't clarify the role of the number of iteration for achieving good performances.

Correctness: Both the mathematical results and the experimental evaluation seems correct.

Clarity: The paper is well structured and easy to follow. Both the theoretical parts and the experimental parts describes well the obtained results.

Relation to Prior Work: In general the paper clearly discusses the relation with previous works. One exception is that the paper doesn't discuss previous work on differentially private ordinary least squares, e.g. [1]. Since this method can also be used for private mean estimation, it should be discussed. [1] Or Sheffet. Differentially private ordinary least squares. In International Conference on Machine Learning, pages 3105–3114, 2017.

Reproducibility: Yes

Additional Feedback: I think that showing the algorithm for covariance estimation (at least Algorithm MVC) would give a better intuition of what this looks like. While I find the description quite insightful, I still felt the need to see the actual algorithm. I find quite confusing how the number of iterations affect the performance of the algorithm, especially for variance estimation. Can you comment more on this? -------------------------------- Comment after authors' response I thank the authors and the other reviewers for clarifying the relations with KLSU19 and their lower bound, I missed it. I find also clearer the role of the parameter t. I suggest to the authors to further integrate the discussion of t in the paper with their response. My overall impression of the paper hasn't changed, I think this is a good submission deserving acceptance.


Review 3

Summary and Contributions: This paper expands on current DP tools to provide a method of producing accurate (concentrated) DP estimates of the mean and covariance of Gaussian data. The method is based on iteratively updating a point estimate and confidence interval for the mean/covariance, resulting in a (hopefully) smaller CI at each step. After this iterative procedure, a much smaller CI about the mean is obtained. The data is then truncated within a prediction interval based on the error of the current CI and tail bounds for the Gaussian distribution, and calibrated noise is added to the empirical estimate. Ultimately, the main contribution is that the proposed method greatly reduces the dependence on the initial bound on the data, and thus allows for a much smaller magnitude of noise to be added for privacy. The authors demonstrate the significance of their contributions both through asymptotic arguments, as well as through persuasive simulations. They also illustrate their approach through PCA on a real-dataset, which highlights the difference in accuracy on a real-data example.

Strengths: The contributions of this paper are important both theoretically and practically. A weakness of most DP mechanisms at the moment is their dependence on prior information, to accurately bound the sensitivity. This paper provides a method to limit this dependence, which they demonstrate both theoretically and empirically. The gains in practical performance that their method provides could allow for more implementations of DP methods on datasets with smaller sample sizes. While their method focuses on a very specific setting of Gaussian data, it seems that their approach can be modified to adapt to other data distributions as well as other sample statistics.

Weaknesses: The present work is limited by its assumption of Normality. In reality, a practitioner likely does not know the shape of the underlying distribution, and would prefer to make weaker assumptions about the data generating distribution. While the authors claim that the results hold for sub-gaussian data, it would be nice to have a formal statement (perhaps a corollary) and proof justifying this claim. Perhaps the results could apply to unknown distributions by using a version of the Markov inequality in place of specifically Gaussian tail bounds. Extensions to these other settings would flesh out the work and make it more complete.

Correctness: The authors seem to have a clear understanding both of prior DP methods as well as their own contribution. The intuition presented in the paper communicated the strategy well, and intuitively made sense. I read through the Proof of Theorem A.1, and it seems correct. I trust that the other proofs, which are extensions of this simple case are also correct.

Clarity: The submitted manuscript seemed to be a significantly abridged version of the larger paper, which was included in the supplementary material. It is my position that the paper should be substantially modified to allow the manuscript to be a self-contained paper, including all of the main results, algorithms, intuition, and related work. On the other hand, if the longer version attached in the supplement is intended to be the full paper, it may be better for the authors to submit to a journal with less stringent page restrictions. My specific critiques are as follows: 1) The related work is cut short, and the full version can only be found in the long-version of the paper. 2) In Sections 2.1 and 2.2, the theoretical results are included, but there is no discussion of the implications of these results. I would like the authors to discuss how the rates in Thm 2.2 & Thm 2.4 differ from the equations in line (91). It is difficult for the reader to directly compare these rates, as they are communicated differently. If the difference only the role of R/K, or do these quantities interact with the dimesion d or privacy parameter rho as well? 3) The algorithm for the estimation of the covariance is omitted from the manuscript. While I understand the page restrictions, this algorithm is one of the main contributions of the paper, and I think it should be in the main paper. 4) In Sections 3 and 4, it is mentioned that the clipping threshold and the confidence ellipsoids are shrunk more ``aggressively''. It is not explained what these thresholds are chosen to be or how they are decided. I get the impression that the simulations implement the algorithms in a different manner from Fig 2/Thm 2.2 (and the analogous algs for the covariance), but it is not explained in the main paper how the tail bounds are modified, or in what other manner the algorithms are changed to improve finite-sample performance. 5) In general, it makes me uncomfortable that a longer version of the paper is included as the supplementary material. This confuses me as to which paper is the ``real'' one. It is my view that the submitted manuscript should stand alone, and that the supplement should be written as an appendix with technical details, proofs, and additional simulations included there. Ideally, only the reader who is interested in replicating or expanding on the work need read the supplement. On the positive side, as mentioned earlier, the intuition presented in the paper is very well presented. I especially liked lines (50)-(64) and the illustration included there, as well as the beginnings of Section 2.1 and 2.2. These parts of the paper did a good job of distilling the complex mathematics to allow the reader to understand geometrically and intuitively the ideas behind the proposed methods.

Relation to Prior Work: In general, the paper does a good job of discussing and citing prior work related to the problem at hand. The authors also do a comprehensive comparison of their method against other state-of-the-art methods, especially for mean estimation. As mentioned earlier, some of the related work is cut short (included in the long-version of the paper). While I do not prefer this approach, I do not this that this limitation of the related work significantly detracts from the paper.

Reproducibility: Yes

Additional Feedback: Some line-by-line comments. Many of these are nit-picking, or opinions based on my personal writing style. Others are typos or questions of clarification. (11) You mention that the empirical mean and covariance are ``optimal''. In what sense do you mean optimal? Can you not find any citations for this statement? (63) ``meanwith'': typo: should be ``mean with'' (71) ``insensitive to any one sample'': typically the term ``sample'' refers to the entire dataset, whereas in this paper ``sample'' seems to refer to both the dataset, as well as on individual's contribution to the dataset. This line could be changed to ``insensitive to any change in one person's data'' or something like that. (72) Continuing with the previous point, ``if they differ on at most one sample'' could be changed to ``two vectors/datsets are neighboring if they differ in one entry'' page 2, footnote 4: typo: remove the word ``are'' (84) With the assumtion that Sigma >= I, it seems that rescaling will change K as well. It would be nice to clarify this. (91) I believe that the equations are not precise. Probably they are big-O upper bounds? The authors should clarify, or state that the bounds ignore constants if that is the case. (119) & (123) This is the first reference to the algorithms. It was difficult for me to find them at first. Should cite where they are found: Left/right algorithm in Fig 2. (Fig 2): The caption should say roughly what the left alg is and is for, as well as the right. (131) ``none of these points'': Is this actually correct? Or should it say that with high probability all of the points are in the ball? If this is the case, what is the probability? Maybe it is Beta? (138) I am not familiar with Omega_tilde notation. Does it ignore log-factors? Or does it mean that n must grow strictly faster than the inside? A short clarification for those less familiar would be helpful. (187) budget is misspelled. (189) what is the clipping threshold set to? How was it chosen? Do the results of the Theorems still hold with the new threshold? (196) ``trimming parameter .1'': Does it mean that 90% of the data are within the trimming? (212) You advantage SYMQ with twice as many samples. Why? (214-216) The smaller dimension is on the left. Switching the order they are mentioned in the sentence and adding ``respectively'' would make it more clear. (224) You claim that your method works well with n<4d, but 4d=2000 is not shown in the middle figure. (223) You say your method is ``competitive'' against the non-private method. Could you quantify ``competitive''? Including the percent increase in error would help. (234) For non-Gaussian data, is the clipping threshold still based on Gaussian tail bounds? (254) ``aggressive shrinking'': What was the shrinking rate set to, and how was it decided? (284) Why is t=1 not included in Figure 7? It would be a helpful baseline to see how multiple iterates improves over the naive approach. (298) Misspelled ``the'' (321) ``private projection'': What does this mean? I am guessing that the covariance is privately estimated using your approach, the top two principle components are identified, and then the data is projected onto these components. However the result of this projection is not private, as the data is accessed again. If this is what you are doing, it is not clear. If it is not, a description of the procedure would be helpful. (331) Instead of saying you give the ``first practical algorithms for DP estimation of the mean and covariance'', which is fairly vague, you could emphasize that your method has significantly reduced dependence on the prior information. It is this aspect of your approach that makes them practical. (Figures) When I printed the paper, I found the plots to be unreadable in print (especially as my printer is black and white). For increased readability, I think only a maximum of two plots per line can fit. To aid readability in black and white, different symbols for different methods (such as X for t=2, triangle for t=3, o for t=4, etc) would be very helpful. As the paper currently stands, I had to open the document on my computer and zoom in, in order to decifer the plots. ============================================= To summarize my opinion of the paper, I like the main results and think they are important and appropriate for NeurIPS. My issues with the paper are in how the paper communicates the details of the work, and their stylistic decisions. Currently, I am marking the paper as a ``marginal accept'', but if the authors can persuade me that they intend to improve the structure of the paper, I will change my score to a clear ``accept''. I look forward to the authors' rebuttal. ============================================= ---- UPDATE ------ ============================================= After reading the other reviews and the authors rebuttal, my opinion is roughly the same. I think that including additional discussion addressing the points from the rebuttal will strengthen the paper. In particular, discussion on extending the results to non-Normal will increase the applicability of the results. However, my main issue with the paper is still the presentation. As I discussed earlier, I found that the NeurIPS manuscript does not entirely stand alone. While the additional discussion will help, the authors have not persuaded me that they intend to make any substantial changes to make the paper more self-contained. Nevertheless, as I still find the work to be interesting and important, I leave my score as 6:marginally above the acceptance threshold.


Review 4

Summary and Contributions: The paper discusses two algorithms for differentially private estimation of the mean and the covariance matrix for a Gaussian distribution. The paper demonstrates the usefulness of the algorithms as given by the estimation error with respect to the true values of the parameters. The results also show that the proposed algorithms outperform the current state of the art, especially for low sample sizes. NOTE: Read the author's rebuttal. I will change my assessment to marginally above the acceptance threshold.

Strengths: The proposed algorithms are fairly simple yet they are efficient as the numerical simulations shows it. The results show that they outperform the state of the art algorithms and that the error incurred by taking privacy into account is acceptable. The algorithms are analysed at both a theoretical and empirical level.

Weaknesses: One item that is of particular concern to me is the influence of the value of t on the results. For example, in figure 3 we see in the left plot that for t=2 the algorithm offers best results. In figure 6 we see that for isotropic covariance t=3 offers best results, whereas for skewed covariance t=2 offers best results. Do you have any intuition about how to choose t in practice? From what I can see currently it is more of a trial and error approach. You do say that for mean estimation t=10 seems to be a good choice, however is this values 'universal'? Can I use this value no matter the problem I have at hand? And how do I balance between having a good error for the mean, but not such a good error for the covariance matrix (or the other way round) with respect to the choice of t? Another item of concern for me is the split of the privacy budget between the iterations. You say that better results are provided if the last step receives a higher privacy budget than the intermediate ones. Do you have any explanation as to why this is the case? Did you perform any experiments to see how the error varies with respect to the split in the privacy budget? In the paper you give the split that you used, however do you know if this split is the optimum one, or do you know if an optimal choice exists to begin with? The proofs for the theorems could have been more formalized, at least at the writing style level. The discussion that is prior to each theorem should have been placed after the theorem and highlighted as sketch of proof. The statement about a large n in the sketch of the proofs for the theorems is not further dealt with in the paper. What does a large enough n mean in your case? This is an important aspect as one of the claim in the paper is that the proposed algorithms are efficient even for small datasets (small n). One item that you do not discuss in the paper is the possible trade-off between error and computational cost. Take for example the case t=10, what do I gain in terms of computational cost and what do I lose in terms of accuracy if I decide to switch to t=5 instead?

Correctness: From my understanding the claims and the methodology are correct.

Clarity: The paper is mostly clearly written. However, one area that definitely needs improvements is the graphical representations. I had to zoom in to at least 150% on the pdf version to be able to read the figures. If I am to print the pdf then the figures are imposible to read. I understand that there is the limitation with respect to the number of pages, however if there are figures present I expect them to be readable without having to zoom in at a significant level.

Relation to Prior Work: I believe the authors correctly pointed out and related their work to existing contributions.

Reproducibility: Yes

Additional Feedback: What about the case for a non-gaussian non-symmetric distribution? Do you have an idea on the behaviour of the algorithms? Could you please clarify why SYMQ algorithm is advantaged in the experiment in the right plot from figure 3? There are some few typos that are present and that should be corrected: - at the top of page 3 the transpose sign is placed on the wrong paranthesis in the expression of the empirical estimate for the covariance matrix (it is my understanding that the vectors are column vectors, if not please specify) - page 5 'We found that assigning most of the privacy budet ...'

[Author Response · NeurIPS 2020]

We would like to thank all the reviewers for their incredibly thorough reading and many comments. We are encouraged
to see that all agree the paper is on an important problem, well-written, and the method itself is highly effective.

**General Comments**: R1 asks about the theoretical comparison with KLSU19, whose bounds we match. Indeed, it is
impossible to surpass the sample complexity bounds of KLSU19, as they prove matching lower bounds (see also [5]).
That said, our focus is going beyond the theory, to make practical and realizable tools for this setting. Our algorithms are
a novel approach based on shrinking confidence sets, and this new approach was crucial to achieve this goal. KLSU's
mean estimation algorithm is essentially the KV baseline, which we beat by substantial margins. Their covariance
estimation algorithm was designed for theory, and we found it impossible to achieve non-trivial accuracy (line 248). As
such, ours is the first effective algorithm for private covariance estimation.

Several reviewers ask about the role of the parameter $t$, the number of iterations, and R4 asked about splitting the
privacy budget across these rounds. Roughly speaking, a larger $t$ allows for weaker "prior knowledge" (i.e., a larger $R$
or $K$), at the cost of spending more privacy budget before the crucial final iteration. The theoretically principled way to
choose $t$ is $\Theta(\log R)$ or $\Theta(\log K)$. Of course in practice, the picture is not as clear-cut. For mean estimation, we found
choosing $t$ to be rather large (i.e., 10) was robustly effective in all scenarios. For covariance estimation, the effective
choice of $t$ seems to be more setting-dependent. The effect of $t$ is explored in depth in the supplement, throughout,
but with a particular focus on the effect of increasing $R$ and $K$ (bottom of page 11 and 15). As for the privacy budget,
the majority of the privacy budget should be allocated to the final round as it plays a special role in the algorithm – it
provides the point estimate which is returned, while previous rounds only shrink the confidence interval. We explored
several splits and found this was adequate for strong performance, practitioners may tune further for improved accuracy.
We note that, given a $t$ and split of the privacy budget, the width of the confidence intervals for mean estimation are
computable in advance. Therefore, one can optimize over these parameters in advance, we will describe this in the final
version (along with a caveat that this doesn't account for bias introduced from aggressive clipping). We will address
readability of figures in the final version. (In particular, we did not have access to a printer due to stay-at-home policies.)
If anything is unclear beyond this, we would be happy to add more discussion/exploration.

More specific responses follow.

**R1**: We note that $t = 1$ corresponds to the private baseline (commonly known as "Analyze Gauss," see also line 246,
caption of Figure 8), and there are no other effective approaches known.

**R2**: See above for discussion of lower bounds and setting $t$. Sheffet's paper assumes a bound B on the data and pays
linearly in this parameter – as our focus is minimizing the cost when B might be large, Sheffet's method would not be
competitive in our setting. We will add discussion to this effect.

**R3**: Thanks to R3 for the numerous comments, due to space restrictions we address a subset here. We are glad the
reviewer found appreciated the intuition and ideas conveyed in the body.

Our method is not restricted to Normal distributions (in fact, the algorithm, theorem, and proof for the sub-Gaussian
case are unchanged), or even sub-Gaussian ones – all we need is the ability to derive tail bounds or confidence intervals
for the class of interest. Given this, only superficial modifications are required (in particular, changing $\gamma$ and $r'$ in lines 2
and 4 of MVM). The algorithms, theorems, and proofs are otherwise effectively the same (but would give different rates
depending on the form of the tail bounds). We will add discussion on this. On other topics: "None of these points" is
indeed a high probability statement. The tilde notation disregards log factors. Details related to clipping thresholds and
aggressive shrinking are documented in our submitted code. For our non-Gaussian experiments, yes, we used the same
Gaussian bounds (testing the effect of model misspecification). Regarding the exclusion of $t = 1$ for clarity in some of
the plots, in those cases, the $t = 1$ line is so far apart from the ones for higher $t$, such that the latter become difficult to
tell apart. On the private projection: the principal component vectors (the "structure" we extracted) are output privately,
the points are plotted only for visualization (releasing them would not be private, as correctly identified).

**R3** and **R4**: SYMQ advantaged: The second half of line 213 says: with the same $n$, SYMQ fails catastrophically
(Figure 4 of supplement). While we could declare superiority at this point, we allow SYMQ twice as many samples to
permit further comparison (which we still outperform).

**R4**: Style comments: We respectfully prefer the current structure, as we believe the theorem statements placed upfront
would be confusing or uninformative. This style of argument leading to a theorem is common in mathematical prose.
Regarding large enough $n$, this refers to the bound in the sample complexity. Details of how this bound is applied are
spelled out in the proof in the supplement.

Computation: The running time is linear in $t$, we can add a comment to this effect. Values of $t$ which are too small will
result in poor accuracy, see theorem statements and Figure 4 of the supplement.

Non-Gaussian non-symmetric distributions: See first comment to R3.

[Meta-Review · NeurIPS 2020]

All four reviewers support acceptance of this paper. They agree that the paper makes both empirical and theoretical progress on the problem of mean and covariance estimation under differential privacy. I therefore recommend accept. In the camera ready, the authors should include discussion of the t parameter of their algorithm (i.e., how to set it to get good performance, and clarify that t=1 is the Analyze Gauss Baseline), mention generalizations beyond sub-Gaussian distributions, and clarify the differences to Kamath, Li, Singhal, Ullman [24].